# Age and learning shapes sound representations in auditory cortex during adolescence

**Benedikt Praegel[1,2], Feng Chen[3,4], Adria Dym[1], Amichai Lavi-Rudel[1], Shaul Druckmann[3,5], Adi Mizrahi[1,2]***

[1]The Edmond and Lily Safra Center for Brain Sciences, Jerusalem, Israel; [2]Department of Neurobiology, The Institute of Life Sciences, The Hebrew University of Jerusalem, Jerusalem, Israel; [3]Department of Neurobiology, Stanford University, Stanford, United States; [4]Department of Applied Physics, Stanford University, Stanford, United States; [5]Department of Psychiatry and Behavioral Sciences, Stanford University, Stanford, United States

## eLife Assessment

This **important** study suggests that adolescent mice exhibit less accuracy than adult mice in a sound discrimination task when the sound frequencies are very similar. The evidence supporting this observation is **solid** and suggests that it arises from cognitive control differences between adolescent and adult mice. The adolescent period is largely understudied, despite its contribution to shaping the adult brain, which makes this study interesting for a broad range of neuroscientists.

**\*For correspondence:**
mizrahi.adi@mail.huji.ac.il

**Competing interest:** The authors declare that no competing interests exist.

**Abstract** Adolescence is a developmental period characterized by heightened plasticity. Yet, how ongoing development affects sensory processing and cognitive function is unclear. We investigated how adolescent (postnatal day 20–42) and adult (postnatal day 60–82) mice differ in performance on a pure tone Go/No-Go auditory discrimination task of varying difficulty. Using dense electrophysiological recordings, we measured spiking activity at single neuron resolution in the auditory cortex while mice were engaged in the task. As compared to adults, adolescent mice showed lower auditory discrimination performance in a difficult task. This difference in performance was due to higher response variability and weaker cognitive control expressed as higher lick bias. Adolescent and adult neuronal responses differed only slightly in representations of pure tones when measured outside the context of learning and the task. However, cortical representations after learning within the context of the task were markedly different. We found differences in stimulus- and choice-related activity at the single neuron level representations, as well as lower population-level decoding of the difficult task in adolescents. Overall, cortical decoding in adolescents was lower and slower, especially for difficult sound discrimination, reflecting immature cortical representations of sounds and choices. Notably, we found age-related differences, which were more pronounced after learning, reflecting the combined impact of age and learning. Our findings highlight distinct neurophysiological and behavioral profiles in adolescence, underscoring the ongoing development of cognitive control mechanisms and cortical plasticity during this sensitive developmental period.

## Introduction

Adolescence is a developmental stage characterized by the continued refinement of perception and cognition. During this period, the brain is highly sensitive to environmental influences, making it

particularly vulnerable to both negative experiences, such as substance abuse, and positive influences, such as supportive relationships (*Hoskins, 2014*; *Hawkins et al., 1992*). While the adolescent brain exhibits greater plasticity compared to adults (*Fuhrmann et al., 2015*), the extent to which specific sensory and cognitive traits are fully developed versus those that remain malleable is still debated. For instance, some studies on sensory perception have demonstrated that adolescent learning is slower and more variable (*Caras and Sanes, 2019*; *Huyck and Wright, 2011*), while others have found that adolescents learn faster in novel contexts, such as reversal learning (*Johnson and Wilbrecht, 2011*). Certain traits, such as cognitive flexibility, are more pronounced in adolescents compared to adults, whereas other behaviors, such as impulsive control, are still immature (*Romer, 2010*). Furthermore, results across studies that target sensory or cognitive traits are often inconsistent (*Wilbrecht and Davidow, 2024*). Therefore, understanding the degree to which sensory and cognitive functions have matured, and the underlying brain mechanisms that support them, remain an ongoing challenge.

A key concept in brain development is that of critical periods (CPs), which are specific time windows during which the brain's neural circuits are highly plastic and particularly sensitive to external experiences (*Hensch, 2005*; *Hensch, 2004*). The early postnatal development of sensory cortices provides classical examples of this phenomenon. In mice, neurons in the visual cortex undergo several CPs such as those for orientation selectivity, direction selectivity, and ocular dominance. During the CP, which occurs in different developmental time windows for different functional attributes, visual experience shapes the specific neuronal attribute to a long-term state (e.g. ocular dominance reaches its adult form, *Hensch, 2005*; *Reh et al., 2020*). The CPs for simple neuronal features in primary sensory cortices are typically complete by the onset of puberty. In the auditory cortex (ACx) of mice, the CP for pure tone processing begins as early as postnatal day 12 (P12), shortly after the ear canals open, and closes by ~P15 (*Kral, 2013*; *Barkat et al., 2011*). Thus, the closure of the auditory CP to pure tones occurs before adolescence begins (*Sun et al., 2010*; *Zhang et al., 2001*). Yet, while some studies have found that simple neuronal properties in the ACx (like response properties to pure tones) are nearly mature by adolescence, others report continued maturation well beyond this developmental stage (*Bhumika et al., 2020*; *Nakamura et al., 2020*). For example, previous studies found that auditory learning and behavioral performance in auditory discrimination tasks involving amplitude modulation detection, as well as temporal interval discrimination tasks remain highly constrained during adolescence (*Caras and Sanes, 2019*; *Huyck and Wright, 2011*). In those tasks, ACx neurons in adolescents exhibit high neuronal variability and lower tone sensitivity as compared to adults (*Caras and Sanes, 2019*).

The perspective shift from viewing development as involving a set of discrete CPs to a broader view of continuous development with multiple, overlapping, CPs that affect each other, calls for studies to test the extent of functional modulation in neuronal features after their CP 'closes.' Thus, although the CP for pure tone representation in the ACx ends by ~P15, we hypothesized that learning of pure tone discrimination would remain malleable in adolescence, because other perceptual features and particularly other cognitive features are still developing. Furthermore, it remains unclear how these features are expressed when measured in the context of active behavior. We measured cortical activity in the ACx of adolescent mice engaged in an auditory discrimination task involving pure tones. Our findings show that learning, behavior, and sound representations to pure tones in the ACx are not fully mature during adolescence.

## Results

### Adult mice outperform adolescents on a difficult, but not an easy, auditory discrimination task of pure tones

To study learning and perceptual performance abilities, we trained and tested adolescent and adult mice on an auditory pure-tone discrimination Go/No-Go paradigm. For this purpose, we utilized an automated behavioral platform called the 'Educage' (*Figure 1a*; *Maor et al., 2020*). We trained adolescent (n=15) and adult (n=15) mice from post-natal day 20 (P20)-P37 and P60-P77, respectively (*Figure 1b*). Following a tone association period (3 days; exposure to Go tones when mice enter the drinking port), mice learned to discriminate between two pure tones separated by 1 octave (7.07 kHz vs 14.14 kHz; *Figure 1b–c*, light blue; herein referred to as the 'easy task'). After one week of training on the easy task (P23 to P30 in pre-adolescent mice and P63 to P70 in adult mice) mice were introduced

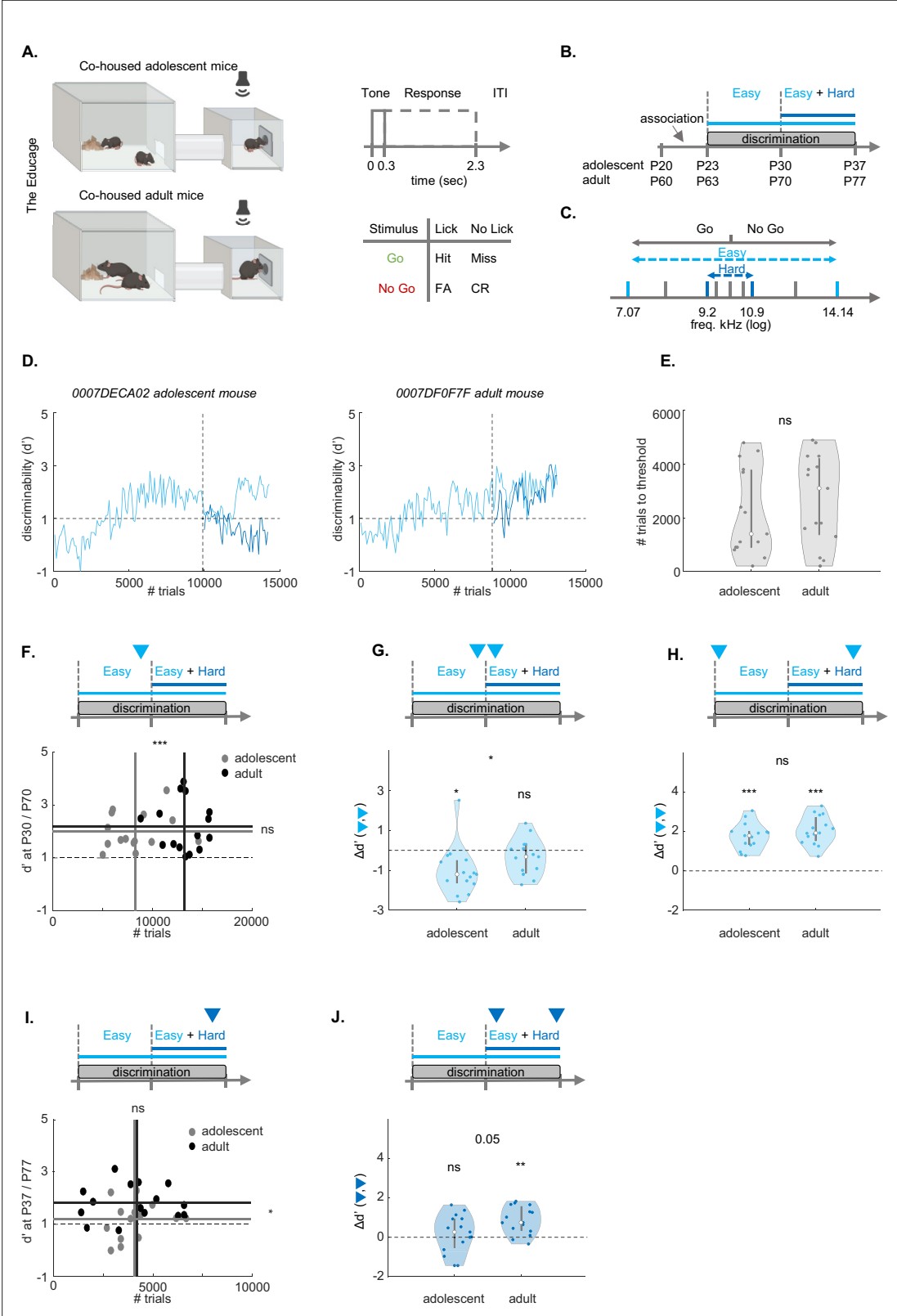

**Figure 1.** Adolescent mice exhibit lower performance during self-initiated auditory learning in the 'Educage'. (**A**) Schematic model of the 'Educage' (left), the trial structure and trial types (FA, false alarm; CR, correct reject). Created with Biorender.com. (**B**) Experimental timeline. Total training time was 21 days (±2). (**C**) The sounds used for training. Light blue: easy task; Dark blue: hard task; Gray: catch-trials. (**D**) Learning curve examples. Adolescent mouse left, (P20–to–P37); adult mouse right, (P60–to–P77). Vertical dashed lines indicate the easy–hard transition. Horizontal line is d'=1. (**E**) Number

*Figure 1 continued on next page*

*Figure 1 continued*

of trials to reach threshold (d' ≥ 1; adolescents, n=15; adults, n=15; z=–0.1659; p=0.8682, two-sample Wilcoxon rank sum test). (**F**) Discriminability (**d'**) of the easy task in adolescent mice at P30 (gray; n=15, d'=2.0001 ± 0.1791; # trials = 8317 ± 712) and adult mice at P70 (black; n=15, d'=2.1986 ± 0.2441; # trials = 13229 ± 514, top: marked by the arrowhead). Dashed lines: mean trials per group (t-stat=–5.6314, df = 28, p=4.9566e-06, two-sample independent t-test, solid vertical line), and mean d' per age group (z=0.2074; p=0.8357, two-sample Wilcoxon rank sum test, solid horizontal line) (**G**) Change in discriminability (Δd') of the easy task before and after the introduction of the hard task (Top: arrowheads; left: adolescent, signed rank = 14, p=0.0067; right: adult, signed rank = 14, p=0.1514, one-sample Wilcoxon signed rank test; Δd' between adult and adolescent mice: z=–2.0739; p=0.0381, two-sample Wilcoxon rank sum test) (**H**) Same as 'G' for the first 100 and last 100 trials of the experiment in the easy task (adolescent signed rank = 120, p=6.1035e-05; adult signed rank = 120, p=6.1035e-05, one-sample Wilcoxon signed rank test; Δd' between adult and adolescent mice: z=–1.0370; p=0.2998, two-sample Wilcoxon rank sum test). (**I**) Same as 'F' for the hard task (adolescent-gray; n=15, d'=1.1895 ± 0.1783; # trials = 4163 ± 297; adult-black; n=15, d'=1.8342 ± 0.1743; # trials = 4102 ± 475; mean trials per group: t-stat=0.1306, df = 28, p=0.8970, two-sample independent t-test, solid vertical line; mean d' per group: z=–2.2398; p=0.0251, two-sample Wilcoxon rank sum test, solid horizontal line). (**J**) Same as 'G' for the hard task. (adolescent signed rank = 73, p=0.4887; adult signed rank = 114, p=8.5449e-04, one-sample Wilcoxon signed rank test; Δd' between adult and adolescent mice: z=–1.9495; p=0.0512, two-sample Wilcoxon rank sum test).

The online version of this article includes the following figure supplement(s) for figure 1:

**Figure supplement 1.** *Behavioral criteria of auditory learning.*

to a second pure-tone pair, now separated by 0.25 octave (9.2 kHz *vs* 10.9 kHz; *Figure 1b–c*, dark blue; herein referred to as the 'hard task'). We completed the experiment after 1 week of learning on the easy and hard tasks simultaneously (i.e. at P37 for adolescents and P77 for adults). During the whole period of training, we also presented mice with low-probability, unrewarded 'probe' trials of pure tones spanning the range of learned stimuli (*Figure 1c*, gray). *Figure 1d* shows example learning curves (of d') from one adolescent and one adult mouse over the 14 day course of tone discrimination. Adolescents and adults learned the procedure of the task similarly well, as measured by the number of trials it took mice to reach a discriminability threshold of d' value ≥1 (*Figure 1e*).

We tested for differences in auditory discrimination by comparing the performance of mice at different epochs along training. First, by the end of the first week of training, all mice performed above the discrimination threshold (*Figure 1f*). Although adolescent mice performed significantly fewer trials during the first week of training (*Figure 1f*; vertical lines: mean # of trials), there was no significant difference between the performance of adult and adolescent mice at the end of the first week (*Figure 1f*; horizontal lines: mean d'; d' calculated from the last 100 trials). Following one week of training on the easy task, we introduced the second pair of pure tones (*Figure 1b*; 'Easy + Hard'). Immediately after introducing the hard pair, the performance of adolescent mice on the easy task dropped, while those of adults remained unaffected (*Figure 1g*). During the second week of training, adolescents regained their high d' values on the easy task, such that by the end of the experiment both groups had similarly high performance on the easy task (*Figure 1h*).

We then compared how mice learned the hard task during the second week while training on both the easy and hard tasks simultaneously. While both groups now performed a similar number of trials (*Figure 1i*, vertical lines: mean # of trials), adult mice outperformed adolescent mice on the hard task (*Figure 1i*, horizontal lines: mean d'), suggesting that adolescents struggled with this harder level of discrimination. Indeed, on average, adolescent mice did not improve on the hard task during the second week of training (*Figure 1j*, 'adolescent'). Adults, on the other hand, improved

**Table 1.** *Behavioral differences between adolescent and adult mice are age-, but not sex-related.*

Fixed effects of age and sex, and the random effects of co-housing in the 'Educage' on the discriminability (mean d' of the last 100 trials of the easy and hard task to avoid pseudo replication) of all mice (Number of observations=30, Fixed effects coefficients = 3, Random effects coefficients = 7, Covariance parameters = 2). Coefficient estimates, STE, T-statistic, degrees of freedom, p-values (adjusted for multiple comparisons with the Bonferroni method) and the lower and upper Confidence Interval (95%). The model includes random effects coefficients of the Cage ID in each group of co-housed mice (7 cages in total; see *methods*, equation 7). Model structure: discriminability(d')~age + sex + (1|cage ID).

| Fixed effects | Estimate | STE | T-statistic | DF | P-value | CI (lower) | CI (upper) |
|---|---|---|---|---|---|---|---|
| Intercept | 1.6001 | 0.2312 | 6.9222 | 27 | **3.8798e-07** | 1.1258 | 2.0744 |
| Age | 0.5994 | 0.2266 | 2.6457 | 27 | 0.0269 | 0.1346 | 1.0643 |
| Sex | –0.0428 | 0.2476 | –0.1729 | 27 | 0.9999 | –0.5509 | 0.4653 |

their performance on the hard task during the second week (*Figure 1j*, 'adult'). Notably, neither group could learn the hard task if it was not preceded by the easy task (*Figure 1*, *Figure 1—figure supplement 1, b*).

Each mouse performed a unique number of trials, as these were initiated by the mice spontaneously. As noted above, pre-adolescent mice performed fewer trials/day as compared to adults but, notably, only during the first week of training (mean ± STE: adolescents, 8317±712 trials, adults 13229±514 trials, t-stat=–5.6314, df = 28, p=4.9566e-06, two-sample independent t-test). Still, this difference raises the question whether the different number of trials affected our conclusions from the first week of training. Thus, to evaluate the possible connection between the number of trials and performance in our data, we carried out several analyses. First, we found that the number of trials and d' were not correlated in either age-group during the first week of discrimination on the easy task (Pearson-r; adolescent: r=0.1880, p=0.5023; adult: r=–0.0730, p=0.7959), nor on the second week of discrimination on both the easy and the hard task (Pearson-r; adolescent: r=0. 2264, p=0. 4171; adult: r=0. 0194, p=0. 9454). Second, we compared performance at three time points along the task with reference to times with shared number of trials: (1) evaluating d' after the minimal number of trials performed by all mice during the first week of training, and (2) evaluating d' at the mean number of trials of each group as a reference. We found no significant differences between adolescents and adults in all comparisons (*Figure 1*, *Figure 1—figure supplement 1c-e*). Additionally, we found no significant differences between males and females, nor among experiments that had different numbers of co-housed mice in the Educage (*Table 1*). Taken together, despite the different absolute number of trials during the first week, we conclude that adolescents and adults learned the easy task equally quickly and effectively. The central difference between the age groups was in their ability to learn the hard task during the second week, when adults outperformed adolescents.

## Adolescents have higher lick bias and higher behavioral variability

We next asked what perceptual and/or cognitive aspect of behavior is different among the age groups. We plotted psychometric curves based on both learned and probe trials from the last 2000 trials of the experiment (*Figure 2a*). The false alarm rate but not the hit rate of adolescent mice was significantly higher as compared to adults (*Figure 2b*). To calculate the decision boundary of mice and their perceptual sensitivity we normalized the psychometric curves of each mouse using a unity-based normalization and sigmoid fitting (*Figure 2c*). We found no differences in the decision boundary or slopes of the curves, suggesting that perceptual sensitivity is not different between the groups (*Figure 2c*). To test whether the differences arise from a cognitive effect (e.g. lick bias), we calculated the decision threshold as the maximal lick bias, also known as the criterion bias (C-bias). C-bias reflects the tendency to respond in a liberal (i.e. negative C-bias) or a conservative (i.e. positive C-bias) manner to the sounds during the task. C-bias was negative for both groups, yet significantly lower in adolescents, suggesting that their cognitive ability to withhold licking is inferior to that of adults (*Figure 2d*).

Given that adolescents and adults show different dynamics in their performance along the task (*Figure 1*), we tested whether the differences we found in the lick bias are consistent at different times along the task. To normalize for different numbers of trials in each mouse, we divided their training episodes into the first and last tertiles during the first week and second week of training separately (*Figure 2e*). The maximal lick bias was significantly higher in adolescents in all but the very first training episode, when mice learned the procedure (*Figure 2f*). These data highlight the general cognitive difficulty of adolescents to withhold licking during the task.

The cognitive sensitivity of adolescents was further evident when their performance dropped right after the task was switched from 'easy only' to 'easy +hard' (*Figure 1f*). The lick bias of adolescent mice became significantly stronger after the introduction of the hard task (*Figure 2g*). In contrast, the lick bias of adult mice remained stable (*Figure 2g*). These data further suggest that this cognitive trait (i.e. to withhold licking) has not yet matured in the young mice. To test if the lick bias was stimulus-related or a result of general impulsivity, we also compared the inter-trial interval (ITI) after different trial outcomes, and observed similar ITIs across groups, except for false alarm trials, where adolescents initiated the trials significantly faster (*Figure 2*, *Figure 2—figure supplement 1*). This, too, suggests that adolescent mice exhibit more impulsive responses following punishments (or alternatively are less impacted by punishments), contributing to their immature lick bias.

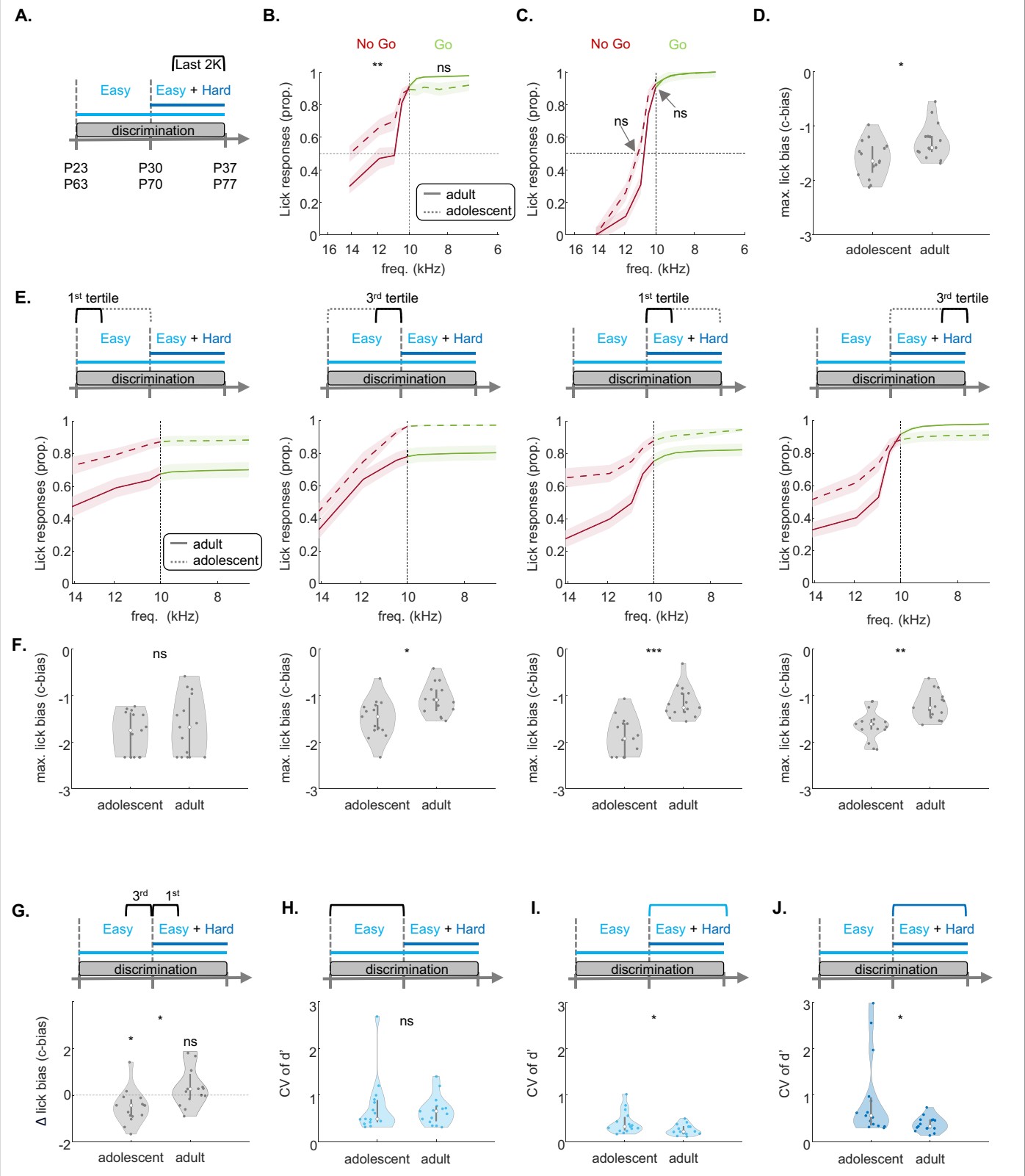

**Figure 2.** Adolescent mice exhibit lower performance in the head-fixed discrimination task. (**A**) Experimental timeline of training followed by recordings. (**B**) Trial structure during the recording. Solid lines indicate the tone period. Dashed lines show the reward or punishment delay (0.6 s), and the response window (2 sec). (**C**) Example session. Licks (gray ticks) and trial outcomes (hit = green, false alarm = yellow, miss = red, and correct reject = blue) across all trials in one recording session. (**D**) Discriminability during training sessions for the easy task (light blue) and hard task (dark blue). (**E**) Change in d'

*Figure 2 continued on next page*

*Figure 2 continued*

after the introduction of the hard task (last 100 trials of the last session of the easy task compared to last 100 trials of the first hard session; adolescent: sign-rank=1, p=0.0625; adult: sign-rank=10, p=0.9998; rank-sum=14, p=0.0381, two-sample Wilcoxon rank sum test). (**F**) Expert d' of the last 100 trials during the last training session of the easy task (rank-sum=21, p=0.1255, two-sample Wilcoxon rank sum test). (**G**) Same as 'F,' but for the hard task (rank-sum=17, p=0.0173, two-sample Wilcoxon rank sum test). (**H**) Behavioral performance (average d' of the easy and the hard task) per mouse during recording sessions for adolescents (n=13, left) and adults (n=14, right; trials per recording: adolescent: 340.5385±45.0650; adult: 431.1429±30.3367; independent t-test, t-statistic=−203.7581, p=0.1116). (**I**) Same as 'H' but only for the first 148 trials. The color bar shows the p-values between the groups. (**J**) Average cumulative licks per trial in adolescents (dashed-line) and adults (solid-line) from −200 ms before tone-onset until the reward or punishment delay, 500 ms after tone-offset. (**K**) Lick latency per trial for adolescent (left) and adult (right) groups during electrophysiological recordings (LME statistics are shown in ***Supplementary file 1***). (**J**) Same as 'K' for the Lick count.

The online version of this article includes the following figure supplement(s) for figure 2:

**Figure supplement 1.** Inter-trial interval after different trial outcomes.

Prior studies have shown that a dominant feature of adolescent behavior is that it is more variable compared to adults (*Caras and Sanes, 2019*). To examine behavioral variability in our data, we calculated the coefficient of variation (CV) of the discriminability (d') for each mouse across training. While there was no significant difference in d' variability when training on the easy task during the first week of training (*Figure 2h*), adult mice showed significantly lower variability during the second week of training while training on the 'easy +hard' versions of the task (*Figure 2i and j*). Thus, adolescent learning and behavior are characterized by both lower response inhibition and higher variability, particularly during the second week when the task involved more challenging discrimination for the younger mice.

## Adult mice outperform adolescent mice on a head-fixed discrimination task

To enable electrophysiological recordings during expert task performance, we trained adolescent (n=5) and adult (n=6) mice on a head-fixed learning paradigm, using a similar protocol as in the Educage (*Figure 3a*). In the head-fixed protocol, water access was limited to training sessions. Head-fixed mice were trained to lick after a 100 ms tone and were rewarded (with water) or punished (by a 2 s white noise) only after a 600 ms delay (*Figure 3b*). Each session was concluded after the mice became satiated and stopped licking. Water supply was limited to 0.0125 ml per day per gram of body weight, and mice that did not consume this amount were compensated after training (see Methods; *Figure 3c* shows an example session from one mouse). The number of trials per training session was significantly lower in adolescents (mean ±STE: adolescents: 410±35, adults: 580±37 t-statistic=−27.9855, p=0.0012, independent two-sample t-test). However, the number of sessions to reach the behavioral threshold (d' ≥ 1 in the easy task) was not significantly different between adolescents and adults (mean ±STE: adolescents: 3.7±0.8, adults: 3.8±0.7; t-statistic = - 0.8531, p=0.1971, independent two-sample t-test, *Figure 3d* shows all learning curves). Similar to learning in the Educage, adolescent performance (but not that of adults) decreased after we introduced the hard task (*Figure 3e*). Another similarity between the head-fixed and Educage versions of the task was that we found no significant differences between adolescents and adults in the performance of the easy task (*Figure 3f*), and that adults outperformed adolescents on the hard discrimination (*Figure 3g*). Also similar to the behavior in the Educage, lick responses of adolescent mice were generally higher (*Figure 3*, *Figure 3—figure supplement 1*), and adolescents exhibited a stronger lick bias (*Figure 3*, *Figure 3—figure supplement 1b*). In line with this result, in the head-fixed task, we found higher impulsiveness in the proportion and the number of licks during the inter-trial intervals after FA trials in adolescents (*Figure 3*, *Figure 3—figure supplement 1c, d*).

To test whether ACx was necessary for expert discrimination in this task, we performed a causal experiment in adult mice. Adult mice (n=3) were injected bilaterally with AAV5-CAMKII-GtACR2-FRED_kv_2.1 into the ACx, and optical fibers were implanted over the injection sites (*Figure 3*, *Figure 3—figure supplement 2a, c*; *Figure 3*, *Figure 3—figure supplement 3*). This viral construct enables optogenetic silencing of excitatory neurons via GtACR2 (*Guillardia theta* Anion Channel-rhodopsin 2), under the control of the CaMKII promoter. It includes a red fluorescent protein for expression verification (FRED) and a soma-targeting motif derived from the Kv2.1 potassium channel to enhance membrane localization.

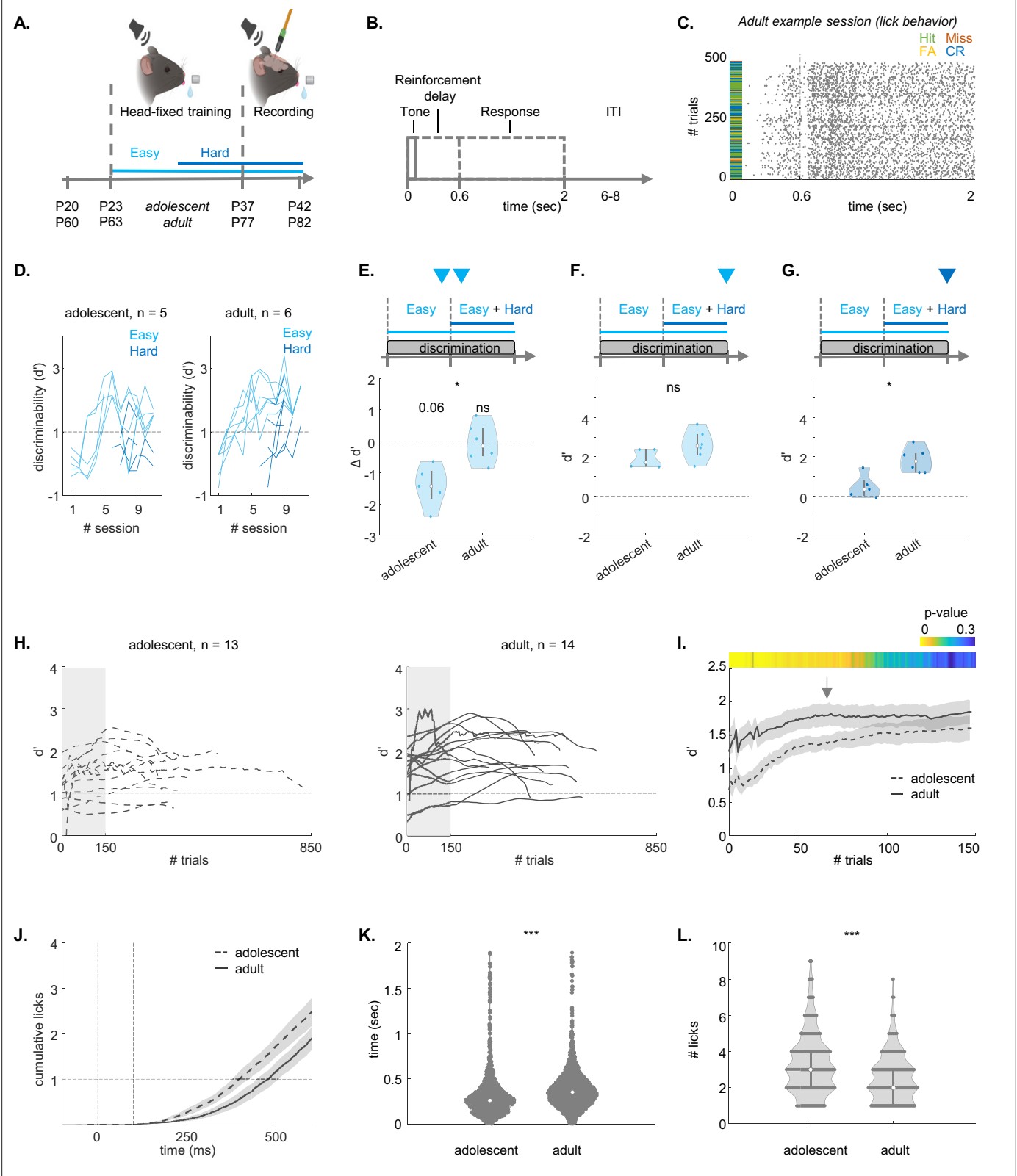

**Figure 3.** Adolescent mice exhibit lower performance in the head-fixed discrimination task. (**A**) Experimental timeline of training followed by recordings. Created with Biorender.com. (**B**) Trial structure during the recording. Solid lines indicate the tone period. Dashed lines show the reward or punishment delay (0.6 s), and the response window (2 s). (**C**) Example session. Licks (gray ticks) and trial outcomes (hit = green, false alarm = yellow, miss = red and correct reject = blue) across all trials in one recording session. (**D**) Discriminability during training sessions for the easy task (light blue) and hard task

*Figure 3 continued on next page*

*Figure 3 continued*

(dark blue). (**E**). Change in d' after the introduction of the hard task (last 100 trials of the last session of the easy task compared to last 100 trials of the first hard session; adolescent: sign-rank=1, p=0.0625; adult: sign-rank=10, p=0.9998; rank-sum=14; p=0.0381, two-sample Wilcoxon rank sum test). (**F**) Expert d' of the last 100 trials during the last training session of the easy task (rank-sum=21; p=0.1255, two-sample Wilcoxon rank sum test). (**G**) Same as 'F,' but for the hard task (rank-sum=17; p=0.0173, two-sample Wilcoxon rank sum test). (**H**) Behavioral performance (average d' of the easy and the hard task) per mouse during recording sessions for adolescents (n=13, left) and adults (n=14, right; trials per recording: adolescent: 340.5385±45.0650; adult: 431.1429±30.3367; independent t-test, t-statistic=−203.7581, p=0.1116). (**I**) Same as 'H' but only for the first 148 trials. The color bar shows the p-values between the groups. (**J**) Average cumulative licks per trial in adolescents (dashed-line) and adults (solid-line) from −200 ms before tone-onset until the reward or punishment delay, 500 ms after tone-offset. (**K**) Lick latency per trial for adolescent (left) and adult (right) groups during electrophysiological recordings (LME statistics are shown in *Supplementary file 1*). (**J**) Same as 'K' for the Lick count.

The online version of this article includes the following figure supplement(s) for figure 3:

**Figure supplement 1.** Lick bias and impulsivity in adolescent and adult mice during head-fixed recordings.

**Figure supplement 2.** Auditory Cortex is necessary for task execution in adult mice.

**Figure supplement 3.** Verification of GtACR2 expression.

**Figure supplement 4.** Adolescent and adult mice performed similarly throughout recordings as well as between the head-fixed configuration and the Educage.

Mice were trained identically on the head-fixed Go/No-Go paradigm as described above. At the end of training, we attached a light source to the implanted fibers to allow optogenetic suppression during several sessions (*Figure 3*, *Figure 3—figure supplement 2a*). In these sessions, light was applied in 50% of trials in a pseudo-random fashion, with light-on trials starting from −50 ms prior to tone onset up to 50 ms after tone offset (*Figure 3*, *Figure 3—figure supplement 2b*). A control group of mice (n=3) were injected with AAV9-CaMKII-dTomato and went through the exact same procedure (*Figure 3*, *Figure 3—figure supplement 2a, c*). We compared the lick responses for both Go and No-Go stimuli under light-on and light-off conditions. We found that lick responses were strongly affected in the GtACR2-injected mice (n=13 sessions), but not in controls (n=8 sessions) (*Figure 3*, *Figure 3—figure supplement 2d*). Light-on conditions only affected the lick responses of the inhibited trial and not the light-off trials (*Figure 3*, *Figure 3—figure supplement 2e*). Discriminability (d') decreased significantly with optogenetic suppression for both easy and hard tasks, although all mice still performed better in the easy task under both conditions (*Figure 3*, *Figure 3—figure supplement 2f*). These results confirm findings by others in similar (though not identical) tasks (*O'Sullivan et al., 2019*; *Ceballo et al., 2019*), suggesting that the ACx is necessary for executing the behavior during the expert stage of this task (but see Discussion for limitations of this experiment).

To measure neural responses during behavioral performance, head-fixed expert mice underwent multiple recording sessions targeting the ACx (adults, n=14 recording sessions from 6 mice, age: P77-P82; adolescents, n=13 recordings from 5 mice, age: P37-P42). During the recordings, the performance of all mice on the easy task was at least d'>1, and the d' on the hard task was smaller for both groups (often smaller than 1; *Figure 3*, *Figure 3—figure supplement 4a*). Performance was not significantly different across the multiple recording sessions in each mouse (*Figure 3*, *Figure 3—figure supplement 4b*). Behavioral performance during recordings was not significantly different from performance in the Educage (*Figure 3*, *Figure 3—figure supplement 4c*). *Figure 3h* shows the evolution of behavioral discriminability (d') during all recordings for adolescents and adults. The performance was heterogeneous across mice as well as within recording-sessions. However, the total number of trials performed was not significantly different between age groups (trials per recording: adolescent: 340.5±45; adult: 431.1±30.3; independent t-test, t-statistic=−203.7581, p=0.1116).

We found additional differences between the behavior of adults and adolescents in the head-restrained configuration. The average adolescent discriminability was lower at the beginning of the session (initial 78 trials) but then rapidly leveled out (*Figure 3i*). *Figure 3j* shows the cumulative lick curves of adults and adolescents. Adolescent mice had shorter lick latencies (*Figure 3k*), and higher lick counts (*Figure 3l*, Supplementary File 1). While the lick latencies were independent of the discriminability (d'), we found a significant interaction effect between lick count and the d' (Supplemetary File 1). Together, these behavioral differences suggest that the weaker response inhibition (here, expressed as lick latency) and higher reward anticipation (here, expressed as lick count) may contribute to the weaker performance of adolescents during the beginning of the recording session.

# Neuronal representations of stimulus- and choice-related activity are immature in adolescents

To compare the activity of neurons in adolescent and adult mice, we recorded spiking activity from the ACx using the high-density Neuropixels-1 probes in expert mice engaged in the task. We targeted the ACx by inserting the probe in a diagonal angle, traversing four auditory regions: Dorsal Auditory Cortex (AUDd), Primary Auditory Cortex (AUDp), ventral auditory cortex (AUDv), and Temporal Association Cortex (Tea; *Figure 4a*; *Figure 4*, *Figure 4—figure supplement 1a, b* show an example voltage trace and one example recording). We recorded multiple times from each mouse and used DiI- or DiO-coatings for different penetrations to verify probe trajectories postmortem. We used the 3D-allen CCF-slice reconstruction platform (https://github.com/cortex-lab/allenCCF; *Shamash et al., 2025*), for high resolution anatomical registration (*Figure 4b*; *Figure 4*, *Figure 4—figure supplement 1c, d*).

We limited our single neuron analysis to well-isolated single units from infragranular layer 5 and layer 6 (recordings were largely restricted to these layers due to the probe angle), as two main projection layers of the cortex, which are key nodes for assessing cortical outputs (*Kanold et al., 2014*). Recorded neurons from other layers and areas were excluded from the analysis. Together, we collected data from 1145 single neurons in 13 recordings of five adolescent mice and 1267 single neurons in 14 recordings of six adult mice (*Supplementary file 2*; an overview of every dataset per figure is detailed in *Supplementary file 9*). We further restricted our analysis to single neurons excited by sounds during the first 150 ms after tone onset. *Figure 4c* shows the normalized firing rate (FR) compared to the normalized lick rate (LR) from –200 ms to 600 ms after the tone onset, before the end of the reinforcement delay (*Figure 4*, *Figure 4—figure supplement 1e* shows the FR in Hz per auditory subregion). FR-PSTHs and LR-PSTHs were not correlated during the first 150 ms (adolescents $r=-0.1515$, $p=0.9878$; adults $r=-0.0638$, $p=0.9942$). In total, we analyzed $n=463$ neurons from adolescent mice and $n=599$ from adults.

In *Supplementary file 2*, we present the number of neurons in our dataset, categorized by region. A detailed overview of the differences in firing properties per auditory region is also provided in *Supplementary files 3 and 4*.

The differences in firing properties between adolescent and adult neurons varied across auditory subregions. AUDp showed differences in all firing properties that we analyzed. In AUDd, adolescent neurons had greater latency to peak, full-width half-maximum, minimal latency, and lower trial-responsiveness. AUDv also showed greater minimal latency and lower lifetime sparseness. TEa neurons of adolescents exhibited greater minimal latency and higher spontaneous FR. While these differences may partially reflect the varying number of neurons recorded per subregion and different functional attributes of each region, we observed similar trends in Cohen's D values (effect sizes) across all four areas, supporting a generally consistent group differences in activity patterns. Consistent with previous work, we also found that firing properties such as minimal latency differed between auditory regions in both age groups (Supplementary File 4; *Feigin et al., 2021*).

To study how adolescents and adults encode task performance, we divided neuronal responses by trial outcome (*Figure 4d* shows one example). Specifically, we assessed neuronal discriminability between stimulus-related and choice-related activity by calculating the area under the curve (AUC) from a receiver operating curve (ROC). Specifically, stimulus-related activity was calculated as the difference between hit and false alarm trials in the easy and the hard task separately (*Figure 4e*, left). Choice-related activity was calculated as the average difference between false-alarm and correct reject trials (*Figure 4e*, left). Miss trials were excluded since we had insufficient trial numbers of this outcome. *Figure 4e* (right) shows the average AUC values over time from an example neuron (same neuron shown in *Figure 4D*), and *Figure 4f* shows the average calculated from all neurons. The vast majority of neurons successfully discriminated (AUC above shuffled data) either or both stimulus- and choice-related activity (adolescent: easy stimuli = 93%, hard stimuli = 93%, choice = 95%; adult: easy stimuli = 97%, hard stimuli = 97%, choice = 95%). The onset-latency of discriminability was significantly slower, maximal discriminability significantly weaker, and the duration of discriminability significantly shorter in adolescent neurons compared to adult neurons (discriminability traces are plotted in *Figure 4f*, and all values from individual neurons and the 3 metrics are plotted together in *Figure 4g*; see *Table 2* for statistics). The four auditory subregions were not significantly different in all three measured metrics (*Figure 4*, *Figure 4—figure*

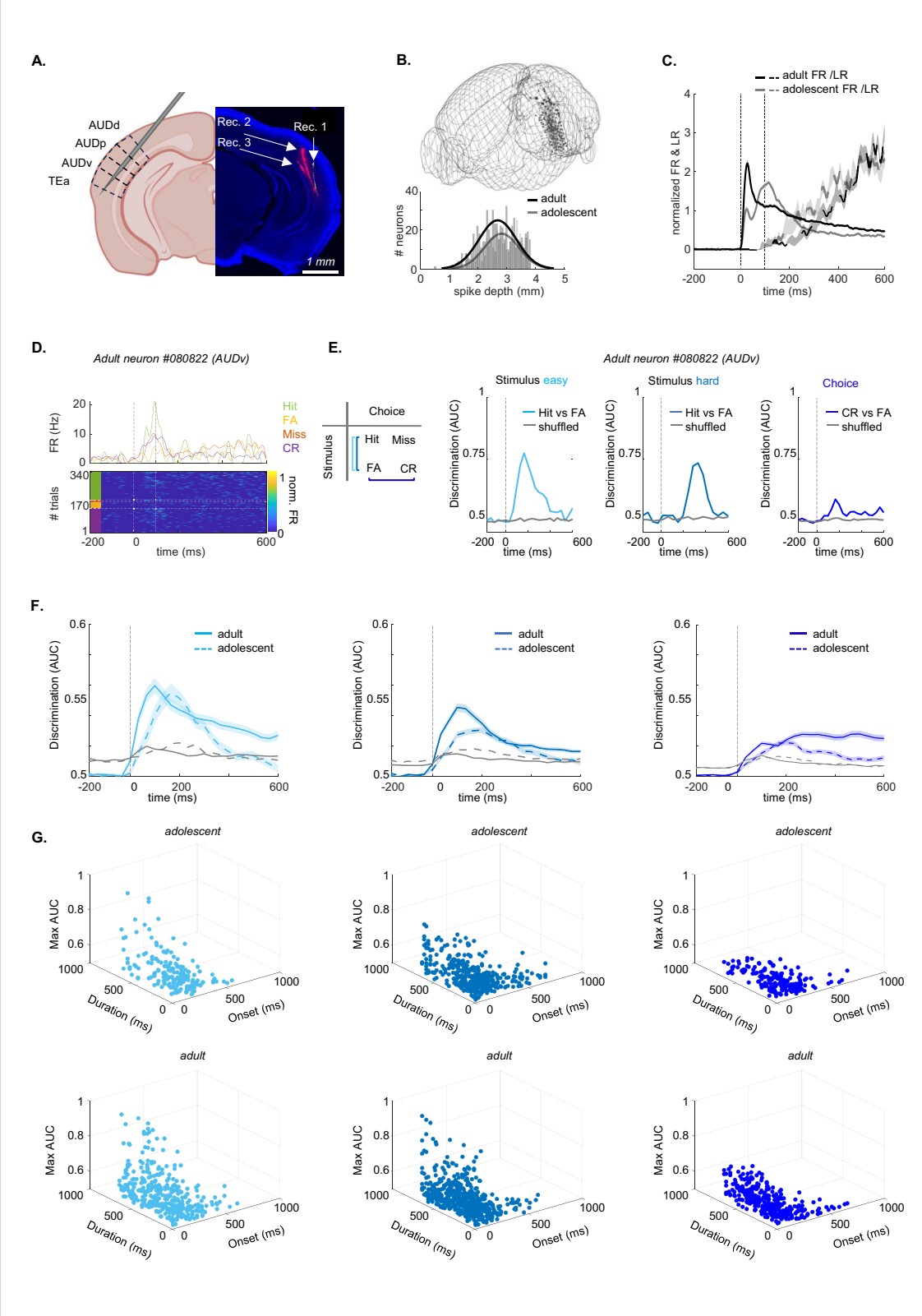

**Figure 4.** Auditory cortex (ACx) neurons in adolescents exhibit lower discriminability in stimulus- and choice- related activity. (**A**) Recordings in ACx when the mouse is engaged in the task, using Neuropixels-1 probes. Left: Recordings were performed in AUDd, AUDp, AUDv, and TEa. Right: Fluorescent micrograph of a coronal brain slice showing the probe tracks of three recordings (red = DiI, yellow = DiO). Created with Biorender.com. (**B**) Top: 3D-Reconstruction of recording sites in adolescents (n=13; gray) and adults (n=14; black). Bottom: distribution of the spike-depth of all excitatory

*Figure 4 continued on next page*

*Figure 4 continued*

tone-responsive L5/6 neurons in adolescents (n=455; gray) and adults (n=607; black). (**C**) Normalized PSTH (FR in Hz) and lick-rate (LR in Hz) from –200 ms to +600 ms after tone-onset in adolescents (gray) and adults (black). (**D**) Spiking activity from one example neuron sorted by trial outcome (hit, miss, false alarm, correct reject). Top: PSTH per trial outcome. Bottom: Heat map of the firing rate (FR) sorted per trial outcome. (**E**) Discriminability values (AUC) over time (from –200 ms to 600 ms after tone onset) for one example neuron (same neuron as in 'D'). AUC values are shown for stimulus-related activity (left: easy task, middle: hard task) and choice-related activity (right). Shuffled distribution in all curves is shown in gray. (**F**) Same as 'E' for all neurons. The curves are average (+- STE) neuronal discriminability of adult neurons (solid line) and adolescent neurons (dashed line), for easy (adolescent neurons = 190, mice = 4, recordings = 7; adult n=358, mice = 4, recordings = 8; left) and hard stimulus-related activity (adolescent n=429; adult n=562, mice = 5, recordings = 9; middle), and choice-related activity (adolescent n=429; adult n=562, mice = 5, recordings = 9; right). (**G**) 3D plots of the onset-latency of discriminability (ms), duration of discriminability (ms), and maximal discriminability (AUC) of all neurons that showed significant discriminability. Left: easy task (adolescent neurons = 178 (93%), mice = 4, recordings = 6; adult n=346 (97%), mice = 4, recordings = 8; left); Center: hard task (adolescent neurons = 399 (93%), mice = 5, recordings = 10; adult n=544 (97%), mice = 6, recordings = 12; middle); Right: choice-related activity (adolescent neurons = 181 (95%), mice = 4, recordings = 9; adult n=339 (95%), mice = 4, recordings = 7; right).

The online version of this article includes the following figure supplement(s) for figure 4:

**Figure supplement 1.** Probe reconstruction and activity profile across auditory cortex (ACx) regions.

**Figure supplement 2.** The neuronal discriminability of stimulus- and choice-related activity is similar across auditory sub-regions.

*supplement 2*). Thus, auditory processing of stimulus- and choice-related activity in the adolescent ACx has not yet reached full maturity.

## Adults exhibit better population decoding of the difficult task compared to adolescents

While individual neurons exhibited weaker discriminability in adolescents than in adults, at a population level, these effects could saturate out. We, therefore, tested the ability of recorded populations to decode trial outcomes. We used a linear discriminant analysis (LDA) decoder to quantify hit versus correct reject trial outcomes. We applied stringent inclusion criteria for the number of simultaneously recorded neurons (see Methods), which restricted our analysis to hit versus correct rejection trials with sufficient trial counts for statistical reliability. We note that this comparison extracts information of the combination of stimulus and choice information indiscriminately (i.e. collectively it includes the full 'task-related' differences). We decoded separately in easy trials and hard trials, based on population activity during the first 200 milliseconds. For the easy task there was no significant difference between decoding accuracy in adolescents and adults (*Figure 5a*). However, decoding accuracy was significantly higher in adults compared to adolescents in hard trials (*Figure 5a*). For both adults and adolescents, decoding accuracy was significantly lower in hard trials compared to easy trials (*Figure 5a*). We performed finer timescale decoding to quantify the onset latency of differences between the activity for hit vs. correct reject trials, defined as the time in which the decoding accuracy first crosses three times the standard deviation away from the baseline (see Methods). We found that the latency was shorter in adults in both the easy and hard trials (*Figure 5b*). In adolescent animals the latency was larger in hard trials (*Figure 5b*) but not in adult animals (*Figure 5b*). We then extended this analysis over a temporal window from –0.5 s to10 s relative to tone onset (*Figure 5c*). In the 200 ms window after the response period (2.0–2.2 s), there is no longer a significant difference in decoding accuracy between adult and adolescent mice as well as between the two task difficulties (*Figure 5*, *Figure 5— figure supplement 1*).

To further quantify the different decoding performance between two tasks difficulties, we plotted the Fisher separation metric ($\frac{\sigma^2_{between}}{\sigma^w_{within}}$, see Methods) for each level of difficulty. While decoding accuracy reflects how well a classifier can distinguish between trial types, the Fisher separation metric specifically quantifies the ratio of between-category variance to within-category variance. The separation ratio of hard over easy tasks was significantly higher in adult mice than in adolescent mice (robust linear regression, see Methods). Separation could have changed due to alterations in the mean or the dispersion around the mean. Testing the dispersion, we found that the variance around the means was not equal across easy and hard trials but rather significantly increased for the hard trials (*Figure 5d–f*). However, there was no significant difference in variance between adolescent and adult mice within the same task (*Figure 5d–f*).

**Table 2.** Neuronal discrimination is later, shorter, and less precise in adolescent neurons. Linear mixed effect models of the neuronal discriminability in adolescence and adulthood per stimulus-related activity in the easy task (Number of observations = 524, Fixed effects coefficients = 2, Random effects coefficients = 10, Covariance parameters = 3), stimulus related activity in the hard task (Number of observations = 943, Fixed effects coefficients = 2, Random effects coefficients = 14), and choice-related activity (Number of observations = 520, Fixed effects coefficients = 2, Random effects coefficients = 10, Covariance parameters = 3). The table shows the fixed effects of the coefficient estimates, STE, T-statistic, degrees of freedom, p-values (corrected for multiple comparisons with Bonferroni-correction) and the upper and lower CI of the effect of age on the onset latency of discrimination, duration of discrimination, and maximal neuronal discrimination (AUC). Each model also included random effect coefficients of each mouse, and recording per mouse. P-values for were adjusted with post-hoc tests using Bonferroni-correction (see *methods,* equation 9). Model structures: onset latency (ms) ~age + (1|Mouse ID) + (1| Recording ID); duration (ms) ~age + (1|Mouse ID) + (1| Recording ID); maximal discriminability (AUC) ~age + (1|Mouse ID) + (1| Recording ID).

**Stimulus easy**

| Fixed effects | Estimate | STE | T-statistic | DF | P-value | CI lower | CI upper |
|---|---|---|---|---|---|---|---|
| Intercept | 163.6104 | 7.9457 | 20.591 | 518 | 6.29587E-69 | 148.0009 | 179.2199 |
| Onset (ms) | −29.7978 | 9.7626 | −3.0523 | 518 | 0.00716101 | −48.9765 | −10.6191 |
| Intercept | 216.2921 | 13.2858 | 16.2799 | 518 | 5.15393E-48 | 190.1919 | 242.3924 |
| Duration (ms) | 63.2599 | 16.3499 | 3.8691 | 518 | 0.000369146 | 31.1401 | 95.3796 |
| Intercept | 0.6035 | 0.007 | 85.6404 | 518 | 0.0001 | 0.5897 | 0.6174 |
| Max AUC | 0.0186 | 0.0049 | 3.7564 | 518 | 0.0001 | 0.0088 | 0.02841 |

Stimulus hard

| Fixed Effects | Estimate | STE | T-Statistic | DF | P-Value | CI lower | CI upper |
|---|---|---|---|---|---|---|---|
| Intercept | 151.1905 | 5.4684 | 27.6479 | 535 | 3.6685E-123 | 140.4588 | 161.9222 |
| Onset (ms) | −35.3357 | 7.1998 | −4.9079 | 535 | 3.25392E-06 | −49.4651 | −21.2063 |
| Intercept | 220.614 | 8.4357 | 26.1525 | 535 | 2.8344E-113 | 204.0591 | 237.1689 |
| Duration (ms) | 55.6268 | 11.1065 | 5.0085 | 535 | 1.9647E-06 | 33.8305 | 77.4231 |
| Intercept | 0.5821 | 0.0034 | 170.2777 | 535 | 0.0001 | 0.5754 | 0.5888 |
| Max AUC | 0.02013 | 0.0048 | 4.1119 | 535 | 4.5413e-05 | 0.0105 | 0.0297 |

Choice

| Fixed Effects | Estimate | STE | T-Statistic | DF | P-Value | CI lower | CI upper |
|---|---|---|---|---|---|---|---|
| Intercept | 174.5856 | 8.259 | 21.139 | 518 | 1.6732E-71 | 158.3605 | 190.8108 |
| Onset (ms) | −24.3644 | 10.2288 | −2.3819 | 518 | 0.052746707 | −44.4595 | −4.2693 |
| Intercept | 228.1768 | 12.92 | 17.6607 | 518 | 1.64628E-54 | 202.7947 | 253.5589 |
| Duration (ms) | 82.0002 | 16.0017 | 5.1245 | 518 | 1.26556E-06 | 50.564 | 113.4363 |
| Intercept | 0.5491 | 0.0027 | 200.1358 | 518 | 0.0001 | 0.5437 | 0.5545 |
| Max AUC | 0.01660 | 0.0034 | 4.8959 | 518 | 3.92843E-06 | 0.01 | 0.0233 |

## Both age and learning affect cortical plasticity in mice engaged in the task

We next studied how learning contributes to cortical plasticity in the different age groups. To separate between age- and learning-related effects, we recorded from the ACx of two new groups of mice (adolescents and adults that are age-matched to expert mice) that are novice on the task (*Figure 6a*, novice P37-P42, recordings = 6; novice P77-P82, recordings = 6, n=3 mice per group). We collected

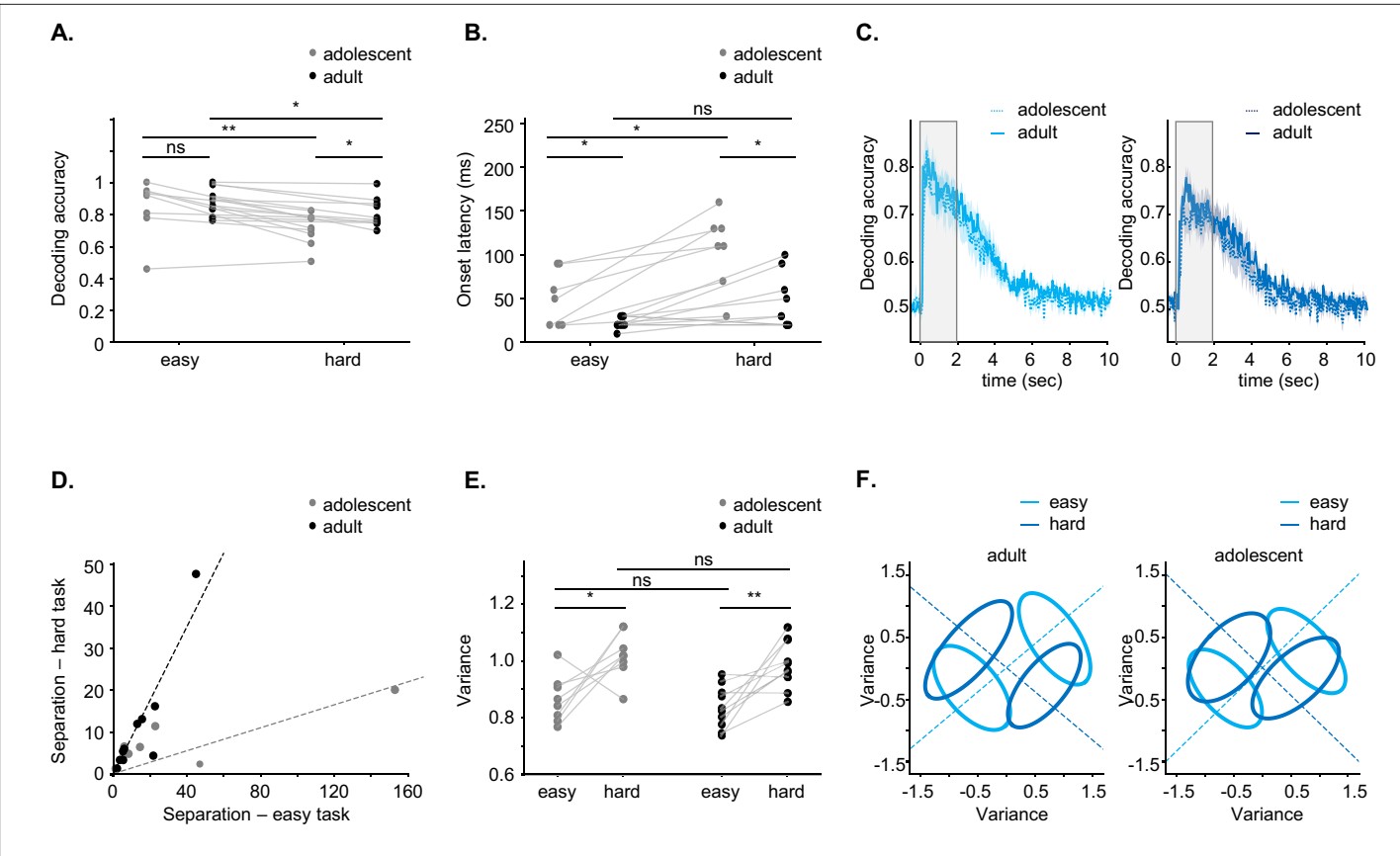

**Figure 5.** Decoding in adult neuronal populations outperforms decoding in adolescents. (**A**) Decoding accuracy for the first 200 ms across all recordings in both adults (black) and adolescents (gray) for the easy task (adolescents compared to adults, p=0.5000, Student's t-test) and the hard task (adolescents compared to adults, p=0.0300, Student's t-test). Decoding is better in the easy task for both age groups (adults: p=0.0030; adolescents: p=0.01, paired t-test). (**B**) Decoding latency for all recordings in the easy task (p=0.0200, Student's t-test) and the hard task (p=0.0030, Student's t-test), as well as compared between age groups (easy task, p=0.05400, paired t-test; hard task: p=0.0100). (**C**) Decoding accuracy over a time window from –0.5 s to 10 s (the response window highlighted in the gray) for the easy task (left) and the hard task (right). (**D**) Linear discriminant analysis (LDA) separation for easy and hard tasks. Lines represent robust linear regression fits without intercept (Huber loss; robust linear regression, p=0.0001) (**E**) Single trial variance for easy and hard tasks in adolescent and adult recordings (adults: p=0.0040; adolescents: p=0.0300, paired t-test; easy task: p=0.4500; hard task: p=0.4100, Student's t-test). (**F**) Visualization of population representations for the stimuli in easy and hard tasks. Dotted lines indicate decoding dimensions, and ellipses represent the covariance of the representations.

The online version of this article includes the following figure supplement(s) for figure 5:

**Figure supplement 1.** Decoding accuracy of the first 200 ms after the response window.

data from 130 tone-responsive (by excitation) neurons in novice adolescents and 186 tone-responsive (by excitation) neurons in novice adults (**Figure 6b**;**Supplementary file 2**; **Figure 6c**) shows the normalized average FR PSTH together with the LR PSTH for novice mice. Notably, novice mice did not discriminate between sounds (**Figure 6—figure supplement 1**). Similar to expert mice (**Supplementary file 3**), we found diverse firing properties across auditory subregions in novice mice (**Supplementary file 5**). Also similar to expert mice, several basic firing properties were different across regions in both novice adolescents and novice adults (**Supplementary file 6**).

To reveal learning-related differences, we compared neuronal responses in novice versus expert mice in both age-groups. We computed single-neuron AUCs for Go and No-Go trials (here, we did not distinguish between stimulus and choice and computed AUCs for all trial outcomes, since expert mice mostly performed CR and novice mice mostly performed FA to No-Go tones), in both easy and hard tasks (**Figure 6d–g**). The onset-latency for AUC discrimination was significantly different between adolescents and adults (shorter in adults), but not between novice and expert mice, nor between the easy and the hard task (**Table 3**). The maximal AUC discriminability was higher in adults compared to adolescents. It was also higher in experts than in novice, and in the easy compared to the hard task

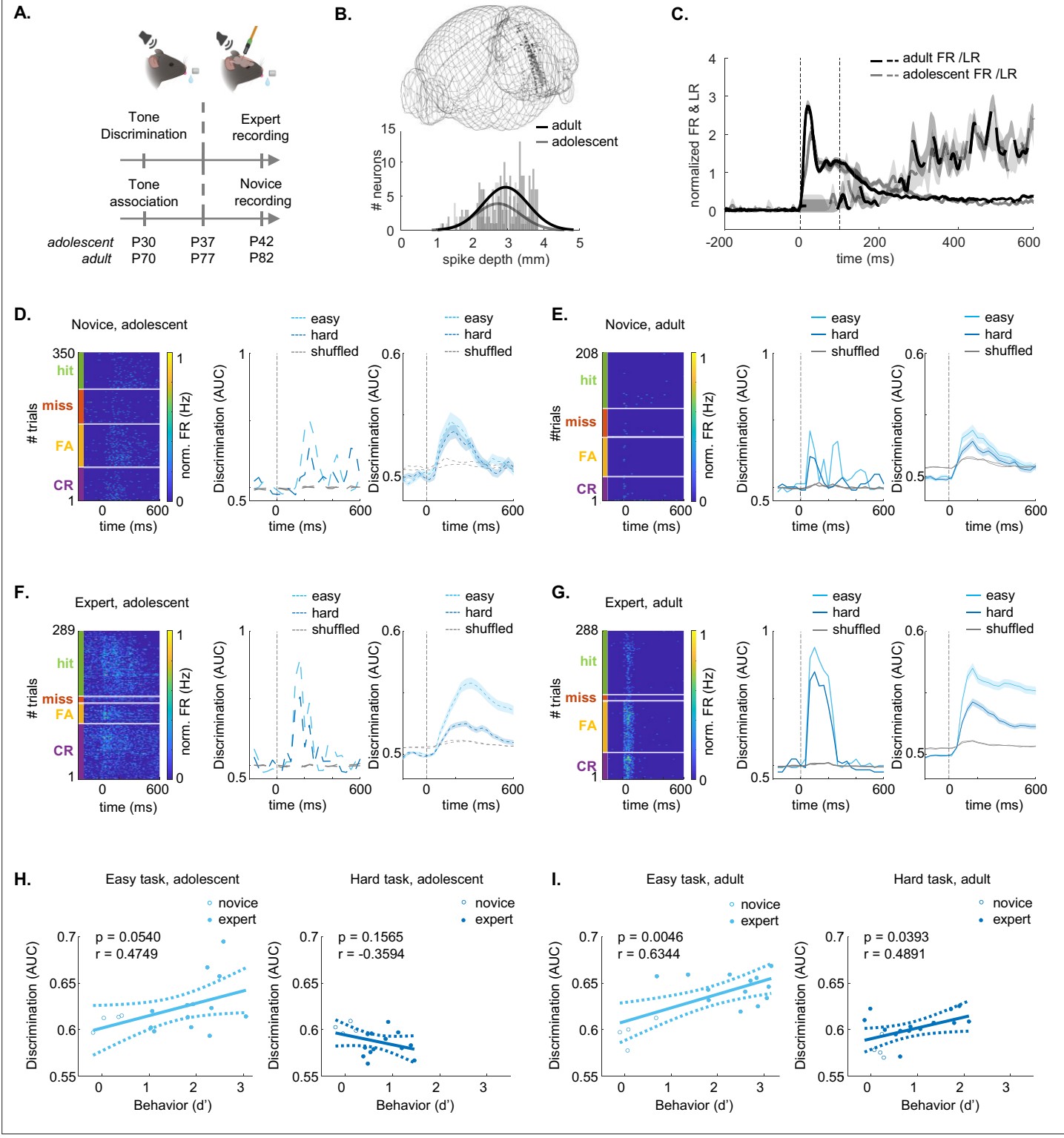

**Figure 6.** Cortical activity during behavior reflects both age- and learning-induced effects. (**A**) Training and recording schedule for novice mice, compared to expert mice. Created with Biorender.com. (**B**) 3D-Reconstruction of recording sites in novice adolescent (n=6; gray) and novice adult (n=6; black) mice. Bottom: spike-depth of excitatory tone-responsive L5/6 adolescent (n=130; gray) and adult (n=186; black) neurons. (**C**) Normalized FR and lick rate (LR) PSTH from −200 ms to 600 ms after tone-onset in adolescents (gray) and adults (black). Average +-sem. (**D**) Single neuron data from novice adolescent mice. Left: Heat map of the firing rate (FR) per trial from one example neuron sorted by trial outcomes. Center: the AUC of the neuron from the left for the easy and hard stimulus pairs (light and dark blue, respectively). Right: Average (+-SEM) AUC of all neurons in the novice group (n=140

*Figure 6 continued on next page*

*Figure 6 continued*

neurons). (**E–G**) Same as 'D' for novice adult (n=186 neurons), expert adolescents (n=455 neurons; Easy vs hard), and expert adults (n=604 neurons; Easy vs hard.). (**H**) Linear regression analysis between the average AUC per recording and the behavioral d' during the recording (the correlation and p-values are indicated for each plot). (**I**) Same as 'I' for adult mice.

The online version of this article includes the following figure supplement(s) for figure 6:

**Figure supplement 1.** Behavioral performance of novice mice.

**Figure supplement 2.** The neuronal discriminability of Easy and Hard Go and No-Go are distributed similar across auditory sub-regions in adolescent and adult novice and expert mice.

(*Table 3*). Similarly, the duration of discriminability was longer in adults and after learning (*Table 3*). There was a significant interaction between age and learning, as well learning and task difficulty in the maximal discriminability (*Table 3*). The onset-latency, duration of discriminability, and maximal discriminability, were not significantly different among the AUDd, AUDp, AUDv, and TEa in the four experimental groups (*Figure 6*, *Figure 6—figure supplement 2*).

To compare neuronal discriminability and behavior more explicitly, we analyzed the correlations between neuronal decoding and behavioral performance using linear regression. In the easy task, both age groups showed high correlation between neuronal discriminability and behavioral discriminability (*Figure 6h and i*; light blue). In the hard task, only adult neurons had significant correlations between neuronal discriminability and behavioral discriminability (*Figure 6h and i*; dark blue).

## Effects of age and learning on tuning properties in passively listening mice

Auditory learning has been shown to induce changes in the tuning properties of sounds in the ACx (*Maor et al., 2020*; *O'Sullivan et al., 2019*; *Kanold et al., 2014*; *Bao et al., 2013*; *Bao et al., 2004*; *Han et al., 2007*; *Anbuhl et al., 2022*; *Jaramillo and Zador, 2011*; *Kato et al., 2015*). To study whether learning-induced plasticity in ACx is distinct in adolescents and adults, we recorded from the same mice described above under passive listening conditions by simply extending the recording following the engaged session (*Figure 7a*; Adolescents recordings = 4; Adults recordings = 4). These data were compared to data from a group of novice mice, recorded under passive listening conditions following their own engaged session (; novice adolescent recordings = 6; novice adult recordings = 6).

We characterized the response profile of ACx neurons using a 'frequency-response area' protocol composed of twenty pure tones (4–40 kHz, spaced at 0.1661 octaves steps), each played for 100 ms at five different attenuations (72–32 dB SPL). We limited our analysis to significant excitatory units only, determined as being auditory responsive in a window from tone onset to 50 ms after tone offset at 62 dB SPL. We then focused our tuning analysis on responses to 62 dB SPL because this sound level was close to the intensity used during the behavioral task (72 dB SPL), while still avoiding potential ceiling effects observed at higher SPLs in recordings of mice not engaged in the task.

Our dataset included four groups. The two adolescent groups were: (1) adolescent novices, and (2) adolescent experts, with 92 and 80 neurons, respectively. The two adult groups were: (3) adult novices, and (4) adult experts, with 123 and 84 neurons, respectively. Neurons in all groups responded to pure tone as expected from previous studies in adult mice, including classical V-shaped frequency response areas (*Feigin et al., 2021*; *Rothschild et al., 2010*; *Cohen and Mizrahi, 2015*; *Cohen et al., 2011*; *Maor et al., 2016*). A representative example from a responsive neuron of an adolescent expert and one from an adult expert are shown in *Figure 7b and c*. To assess differences in pure-tone responses between auditory subregions, we quantified the firing properties of AUDp versus AUDv *Supplementary file 8*; TEa and AUDd were excluded since fewer than 20 units responded to the FRA protocol after task engagement, see (*Supplementary file 7*). We found that the latency to peak was slower in adolescent AUDp. The FWHM and minimal latency were slower in the adolescents in both AUDp and AUDv. In the adolescent AUDv, additional parameters (like spontaneous- & evoked-FR, trial-responsiveness, lifetime sparseness), were different than those in adults. There were no significant differences between AUDp and AUDv within age groups (*Supplementary file 8*).

Next, we tested the learning-related effects among the different groups. The peak of frequency tuning was heterogeneous across the frequency range with no clear trends of learning or age (*Figure 7d*). To quantify these responses, we calculated tuning width (*Figure 7e*), population sparseness

**Table 3.** The effect of age, learning and task difficulty on the latency, duration, and ability to discriminate tones in ACx neurons. Linear mixed effect models of the effect of age, learning, and task difficulty on onset-latency of discrimination, duration of discrimination and maximal discriminability (Number of observations = 2590,, Fixed effects coefficients = 8, Random effects coefficients = 20, Covariance parameters = 3). The table shows the fixed effects of the coefficient estimates, STE, T-statistic, degrees of freedom, p-values (corrected for multiple comparisons with Bonferroni-correction), and the upper and lower CI. The model also includes random effects coefficients of each mouse (adolescent novice = 3, adult novice = 3, adolescent expert = 5, adult expert = 6) and recording per mouse (n=3). P-values were adjusted with post-hoc tests using Bonferroni correction (see *methods*, equation 10). Model structures: onset latency (ms) ~age* learning * difficulty + (1|Mouse ID) + (1| Recording ID); duration (ms) ~age* learning * difficulty + (1|Mouse ID) + (1| Recording ID); maximal discriminability (AUC) ~age* learning * difficulty + (1|Mouse ID) + (1| Recording ID).

**Fixed Effects**

| Onset latency (ms) | Estimate | STE | T-statistic | DF | P-value | CI lower | CI upper |
| --- | --- | --- | --- | --- | --- | --- | --- |
| Intercept | 92.6519 | 3.8909 | 23.8125 | 2582 | **5.4024E-113** | 85.0223 | 100.2815 |
| Age | −22.6378 | 5.0624 | −4.4718 | 2582 | **2.42782E-05** | −32.5645 | −12.7111 |
| Learning | 5.6206 | 8.7235 | 0.6443 | 2582 | 0.9999 | −11.4851 | 22.7263 |
| Task difficulty | 0.8813 | 5.4571 | 0.1615 | 2582 | 0.9999 | −9.8194 | 11.5819 |
| Age- Learning | −6.8831 | 11.1114 | −0.6195 | 2582 | 0.9999 | −28.6713 | 14.905 |
| Age - Difficulty | 0.2461 | 7.2185 | 0.0341 | 2582 | 0.9999 | −13.9085 | 14.4006 |
| Learning - Difficulty | −2.5145 | 12.3257 | −0.204 | 2582 | 0.9999 | −26.6837 | 21.6546 |
| Age-Learning-Difficulty | 4.4526 | 15.7691 | 0.2824 | 2582 | 0.9999 | −26.4688 | 35.374 |

Fixed Effects

| Duration of Discrimination(ms) | Estimate | STE | T-Statistic | DF | P-Value | CI lower | CI upper |
| --- | --- | --- | --- | --- | --- | --- | --- |
| Intercept | 299.7098 | 7.8305 | 38.2747 | 2582 | **8.4469E-254** | 284.3551 | 315.0645 |
| Age | 63.2136 | 9.2513 | 6.833 | 2582 | **3.1017E-11** | 45.0729 | 81.3544 |
| Learning | −88.8699 | 15.9421 | −5.5745 | 2582 | **8.21813E-08** | −120.1306 | −57.6092 |
| Task difficulty | −53.9606 | 9.9745 | −5.4098 | 2582 | **2.06623E-07** | −73.5194 | −34.4017 |
| Age- Learning | −30.6998 | 20.3155 | −1.5112 | 2582 | 0.3926 | −70.5361 | 9.1365 |
| Age - Difficulty | −18.8772 | 13.194 | −1.4307 | 2582 | 0.4579 | −44.749 | 6.9947 |
| Learning - Difficulty | 53.8178 | 22.5289 | 2.3888 | 2582 | 0.0509 | 9.6414 | 97.9943 |
| Age-Learning-Difficulty | 12.9698 | 28.823 | 0.45 | 2582 | 0.9999 | −43.5488 | 69.4884 |

Fixed Effects

| Maximal Discrimination (AUC) | Estimate | STE | T-statistic | DF | P-value | CI lower | CI upper |
| --- | --- | --- | --- | --- | --- | --- | --- |
| Intercept | 0.6278 | 0.0042 | 148.636 | 2582 | **0.0001** | 0.6195 | 0.6361 |
| Age | 0.024 | 0.0056 | 4.2953 | 2582 | **5.42797E-05** | 0.0131 | 0.035 |
| Learning | −0.0249 | 0.0096 | −2.5859 | 2582 | **0.029303232** | −0.0438 | −0.006 |
| Task difficulty | −0.0438 | 0.006 | −7.24 | 2582 | **1.76807E-12** | −0.0556 | −0.0319 |
| Age- Learning | −0.03 | 0.0123 | −2.4376 | 2582 | **0.0446** | −0.0541 | −0.0059 |
| Age - Difficulty | −0.0023 | 0.008 | −0.2846 | 2582 | 0.9999 | −0.0179 | 0.0134 |
| Learning - Difficulty | 0.0394 | 0.0136 | 2.8893 | 2582 | **0.0116** | 0.0127 | 0.0662 |
| Age-Learning-Difficulty | −0.0094 | 0.0175 | −0.5387 | 2582 | 0.9999 | −0.0437 | 0.0248 |

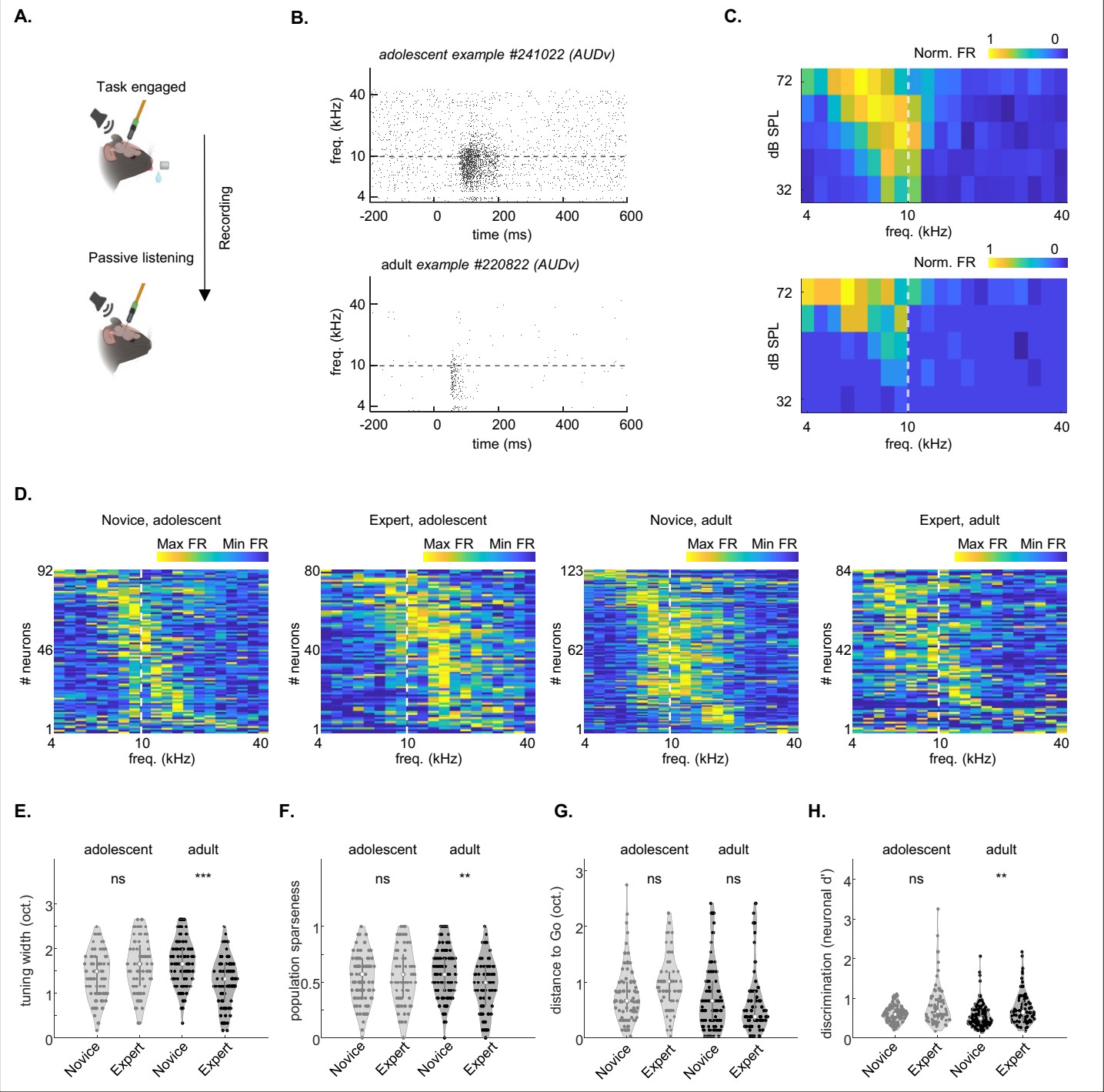

**Figure 7.** Adult mice show greater plasticity after learning. (**A**) Schematic showing that for the passive listening protocol, we continued our recording following the session of the engaged task (i.e. in satiated mice) by removing the waterspout. Created with Biorender.com. (**B**) Example raster plot of a neuron from an adolescent mouse (top) and an adult mouse (bottom). (**C**) Frequency-response areas (FRA's) of the neurons shown in 'B.' (**D**) Distribution of best frequencies in our dataset. Values are normalized firing rates calculated at 62 dB SPL. Matrices are sorted by BF for clarity. Dotted line marks the decision boundary. (**E**) Tuning bandwidth at 62 dB SPL of neurons in adolescents and adults. Side-by-side comparisons of novice versus experts. (adolescents p=0.0882, adults p=0.0001, Kruskal-Willis Test after Tukey-Kramer correction for multiple comparisons). (**F**) Same as E. for Population sparseness (adolescents p=0.9549, adults p=0.0013, Kruskal-Willis Test after Tukey-Kramer correction for multiple comparisons). (**G**) Same as E. for the distance (in octaves) between the best frequency of each neuron to the easy Go-stimulus (adolescents p=0.0816, adults p=0.6391, Kruskal-Willis Test after Tukey-Kramer correction for multiple comparisons). (**H**) Same as E. for the average neuronal d' of frequencies in the learned frequency spectrum (adolescents p=0.1627, adults p=0.0026, Kruskal-Willis Test after Tukey-Kramer correction for multiple comparisons).

*Figure 7 continued on next page*

*Figure 7 continued*

The online version of this article includes the following figure supplement(s) for figure 7:

**Figure supplement 1.** Learning related changes in neuronal tuning properties in primary and secondary auditory cortex.

(*Figure 7f*), the distance between the best frequency of the neurons and the Go frequency of the easy task (*Figure 7g*), and single neuron discriminability (*Figure 7h*). As expected from previous work, learning significantly changed (3 out of 4) neuronal tuning properties in adults (*Figure 7e–h*, 'Adults'). Surprisingly, however, learning had no significant effects on any of these single neuron parameters in the adolescent group (*Figure 7e–h*). To test this at specific brain regions, we analyzed AUDp and AUDv separately, which revealed specificity in learning-induced changes. Learning significantly affected 3 out of the 4 parameters in AUDp of adolescents (*Figure 7*, *Figure 7—figure supplement 1a-d*). However, neurons from AUDv of adolescents showed no learning-induced changes (*Figure 7*, *Figure 7—figure supplement 1e - h*). Notably, adult learning-induced plasticity was stronger than that in adolescence in all parameters besides the 'BF to Go' (*Figure 6—figure supplement 1*). Thus, despite the notion that adolescents' brains may be more plastic than adults, we found generally stronger learning-induced plasticity in basic tuning properties in adults. These results underscore that learning-induced plasticity manifests differently in adolescents and adults.

## Discussion

The term 'stormy adolescence' aptly captures what we know from psychology, neuroscience, and biology—that adolescence is a period of turbulent development, which shapes the individual's future behavior and health (*Sawyer et al., 2012*; *Spear, 2000*). While numerous behavioral phenomena of adolescence have been studied, particularly in humans, the neural underpinnings of these remain largely unknown (*Dahl et al., 2018*). In this study, we focused on (auditory) cortical neurons of the adolescent brain of mice while they were engaged in an auditory discrimination task. We found numerous differences in behavior, single neuron activity, and population encoding between adolescents and adults.

### Adolescence behavior – perceptual and cognitive differences

We utilized the Educage platform to compare between adolescents and adults (*Maor et al., 2020*). The Educage provides an optimal environment for training young mice; its automated nature minimizes human handling, which increases efficiency and reduces to a minimum any interference during the training process. Using the Educage, we found that procedural learning was intact in pre-adolescent mice (P23-P30), and behavioral differences were found only after the mice learned the task during the hard discrimination phase (P30-37). Compared to adults, adolescent mice scored lower on the hard versions of the task (*Figure 1i–j*). Interestingly, our data suggests that the lower performance in the hard task in adolescents was not due to a deficiency in perceptual sensitivity (*Figure 2c*), but rather to a cognitive deficit – namely, lick bias (*Figure 2d*). This result is perhaps not surprising since higher lick bias is a cognitive phenomenon that has been well documented in adolescence as a maturation period of inhibitory control (*Reynolds et al., 2019*). Indeed, additional cognitive control mechanisms are still developing during this age (*Meyer and Bucci, 2016*; *Magis-Weinberg et al., 2019*; *Cohen et al., 2010*).

Another manifestation of weaker cognitive control mechanisms is the noisy behavioral performance of adolescents (*Figure 2h–j*). While this noise may reflect higher plasticity and potentially serve cognitive flexibility (*Johnson and Wilbrecht, 2011*; *Crone and Dahl, 2012*), it poses a disadvantage for the precision and consistency required in our task. This noise may also arise from changes in pubertal hormones, which we did not manipulate (*Kunkhyen et al., 2018*). Notably, few studies have explicitly characterized auditory discrimination of adolescents in rodents; the majority of these have been carried out in gerbils using an amplitude modulation (AM) detection task (*Caras and Sanes, 2019*; *Anbuhl et al., 2022*; *Buran et al., 2014*). Despite the different tasks, many of our age-related findings are aligned with the findings in gerbils, including lower performance in adolescents, and greater variance in adolescent behavior (*Caras and Sanes, 2019*). However, some findings differ. For example, the lick bias which was different in our data, was similar between young and adult gerbils in an AM modulation detection task. This difference may arise from differences in task design, as well as

the severity of the punishment used in the task (here, we used a milder punishment). Here, our data support a view that cognitive factors, and to a lesser extent perception of pure tones per se, are the primary contributors to the poorer performance observed in adolescent mice when performing a pure-tone sound discrimination task.

## Representation of stimulus and choice in ACx still undergoes maturation in adolescence

Motivated by the abovementioned behavioral differences, we tested how single neurons in ACx represented sounds (by comparing Hits vs FA) and choices (by comparing FA to CR). While cortical representations of sounds in ACx could be the basis of perceptual constraints, distinct processing of choices would reflect that more cognitive mechanisms are involved. We found that cortical representations of sounds and choices in mice engaged in the task were still immature as compared to adults, e.g., responses in adolescents were more delayed and less informative (*Figures 4–6*). These differences can be a result of age-related differences, and/or the result of a combined 'age x learning' effect.

We found only modest age-related differences in spiking activity of layer 5/6 neurons between adolescents and adults (*Figure 7*). This limited difference is perhaps not surprising because the CP for tonal organization in ACx is already complete by the age at which we tested the animals i.e., on P37-P42 (*Hensch, 2005*; *Barkat et al., 2011*). In contrast, more pronounced differences emerged from the interaction of 'age x learning,' specifically in mice that learned the task and were actively engaged in the behavior (*Figure 4f*; *Figure 6d–g*). This interaction is evident in the lower single-neuron discriminability (AUC) during the hard task in expert adolescent mice (*Figure 6d* vs. 6 f). While this may seem counterintuitive, it is compatible with the idea that learning can drive complex reorganization in sensory representations. For example, we recently showed that learning leads to distinct and sometimes opposing effects on single-neuron responses with respect to whether the task is perceptually easy or difficult (*Haimson et al., 2024*). It is also important to note that our recordings were restricted to infragranular layers (5/6) and did not sample supragranular (layers 2/3) neurons, which may exhibit distinct learning-related plasticity (*Kanold et al., 2014*). As such, the developmental differences we report may reflect layer-specific properties of the deep output layers.

We speculate that immature feedback to ACx may underlie these differences for several reasons. First, feedback from higher cortical regions to more primary cortices has been proposed as a neural substrate for difficult perceptual discriminations (*Ahissar and Hochstein, 2004*; *Ahissar and Hochstein, 1997*). Second, feedback activity (often referred to as 'top down') is thought to be the key neural pathway involved in cognitive control mechanisms (*Zagha, 2020*). Feedback is typically expressed in the late (>100 ms) phase of single neurons activity, which was significantly different between adults and adolescents across our measured parameters (onset-latency, duration of discriminability, and maximal discriminability; *Figure 4g*, *Table 2*, *Table 3*). The precise source of feedback that contributes to perceptual or cognitive differences remains to be determined, but likely candidates include regions that project to ACx (*Tasaka et al., 2020*; *Tasaka et al., 2023*), such as medial prefrontal cortex (*Delevich et al., 2021*; *Konstantoudaki et al., 2018*), orbitofrontal cortex (*Johnson et al., 2016*), or anterior cingulate (*Nabel et al., 2020*). Testing the causal role of these regions during adolescence will be an important direction for future research.

Cortical coding is performed by populations of neurons, which can efficiently encode stimuli on a trial-by-trial basis (*Druckmann and Chklovskii, 2012*; *Kang and Druckmann, 2020*; *Li et al., 2016*; *Wei et al., 2019*; *Wei et al., 2020*). Population coding, as measured here by decoding accuracy, largely recapitulated the results of the single neuron data. Specifically, cortical populations in adults encoded the different tones better and faster; particularly in the hard task (*Figure 5a–c*). Interestingly, by testing the quality (i.e. separation) of the decoding performance, we revealed a correlation between decoding in the easy and hard tasks in adult (*Figure 5d*). While this result reflects the collapsed information of stimulus and choice, it suggests that the cortical network uses shared representation for decoding the easy and hard tasks, possibly by the same neurons. This is consistent with our recent work showing that indeed the same neurons in ACx respond to the same stimulus differently, depending on if the mouse is engaged in discriminating an easy sound pair or a hard sound pair (*Haimson et al., 2024*). The shallower slopes in adolescence suggest a less efficient decoding scheme that may arise from different populations of neurons encoding task-related information.

Our optogenetic silencing of ACx in expert adult mice resulted in a marked suppression of licking behavior (*Figure 3*, *Figure 3—figure supplement 2*), consistent with the idea that ACx contributes to task performance. However, the strength of the observed effect raises the possibility that the manipulation may have impacted broader aspects of behavior—such as arousal, motivation, or motor readiness—rather than selectively disrupting auditory discrimination. Recent work (*Drieu et al., 2025*) suggests that ACx plays a critical role during learning by supporting higher-order computations but may become less essential for task execution once mice reach expert performance. These findings, together with previous reports (*O'Sullivan et al., 2019*; *Ceballo et al., 2019*), indicate that the necessity of ACx for auditory-guided behavior may depend on task structure, training stage, and the exact nature of the manipulation. The robust decrease in lick rate and performance observed in our experiment may reflect the intensity, specificity, and timing of the silencing. The use of GtACR2, a highly potent silencer, may have further amplified the effect. We cannot rule out the possibility that silencing the output from ACx also disrupted the flow of information to downstream areas, thereby exerting a stronger influence on perception and behavior. Although the injection was targeted to the ACx, off-target effects are also likely to have contributed. Future experiments using graded or temporally specific perturbations will be useful to define more precisely the contribution of ACx across learning and development.

## Learning in the adolescent brain

Cortical representations of pure tones have a critical period (CP) between P12-P15 (*Barkat et al., 2011*; *Bhumika et al., 2020*; *Nakamura et al., 2020*). Indeed, we chose to study animals at P37-P42 on a pure tone discrimination task precisely to avoid the hypersensitivity associated with this CP. But this may be unavoidable, as recent work has shown that ACx also has a later CP for more complex sounds features (*Bhumika et al., 2020*; *Nakamura et al., 2020*). Those late CPs overlap with our training schedule and the underlying biological manifestation of this CP is expected to affect learning-related plasticity in general. The underlying mechanism of the late CP to complex sounds has been associated with dynamic changes of inhibitory circuits (*Takesian et al., 2012*; *Takesian et al., 2010*). Thus, immature inhibitory circuits remain sensitive in adolescents, and could affect how pure tones are represented, despite the closure of the classic tonal CP. Notably, sound discrimination learning changes tonal representations even in adult mice, well after the closure of CP's for both simple and complex sounds (*Maor et al., 2020*; *Bao et al., 2004*; *Haimson et al., 2024*). We hypothesize, therefore, that the differences in performance on more difficult discriminations between adolescents and adults are shaped primarily by age-specific learning-related plasticity, rather than by differences in pure tone representation per se.

Taken together, our findings show that cortical responses in adolescents differ from those in adults. Notably, both groups learned the same pure tone discrimination task under identical training protocols. Although the CPs for pure tones are considered closed by this age, learning to discriminate between tones still varies across developmental stages. It remains unclear how later CPs, such as those shaping responses to amplitude modulation or frequency-modulated sweeps through inhibitory circuits, contribute to these differences. Likewise, the role of ongoing maturation in cognitive control, including the development of feedback pathways, warrants further investigation. Nevertheless, it is evident that developmental processes in the adolescent brain continue to influence how even simple sounds are encoded through learning. The adolescent-specific plasticity observed here directly reflects on how perceptual and cognitive features are integrated within the auditory cortex.

## Inclusion and diversity

We support inclusive, diverse, and equitable conduct of research.

## Methods

### Animals

A total of 47 (37 female and 10 male) C57BL/6 mice were used in this study. Adolescent mice were weaned at postnatal (P) day 20. Adult mice were trained starting from P60. All mice were housed on a 12 hr light/12 hr dark cycle with food and water ad libitum, unless used for behavioral training (see below). All experiments were approved by the Institutional Animal Care and Use Committee (IACUC)

at the Hebrew University of Jerusalem, Israel (Permit Number: NS-21-16448-4, NS-21-16694-4, NS-22-16966-4).

## Surgical procedures

The details for surgical procedures were identical to those used in previous studies from our laboratory (*Maor et al., 2020*; *Feigin et al., 2021*; *Tasaka et al., 2020*; *Gilad et al., 2020*; *Gilday et al., 2023b*; *Shani-Narkiss et al., 2020*; *Gilday and Mizrahi, 2023a*). The relevant procedures are briefly described below.

### RFID Implantation

24 hr before the start of behavioral training in the automated home-cage, mice were implanted with a radio frequency identification (RFID) chip (Trovan, EX). Animals were anesthetized with 2% isoflurane with pure O2 as carrier. RFID chips were implanted under the scruff. After implantation, mice were injected with 0.04 mg/g of 10% Meloxicam solution to prevent infection and alleviate any possible pain from the implantation.

### Head-bar implantation & craniotomy

48 hr before the start of behavioral training on the head-fixed recording setup, mice were anesthetized with 2% Isoflurane, the body temperature maintained at 37 °C and the eyes were covered with 5% Chloramphenicol ointment to prevent from drying. Before removal of the scalp, we applied 4% Lidocaine above the skin and skull area. Afterwards, we glued a custom-made titanium bar at the back of the mouse skull using dental cement (Meta-bond). To enable acute recordings in ACx, we performed a craniotomy on the left hemisphere at 2.5 mm posterior and 4.0 lateral to the bregma. The craniotomy was protected by a pool of dental cement and covered with a silicone elastomer (WPI; Kwik-Cast catalog #KWIK CAST). After the craniotomy, mice were injected with 0.04 mg/g of 10% Meloxicam solution and 0.2 ml of Saline. During the 48 hr of recovery, mice received two further doses of 0.04 mg/g of 10% Meloxicam solution. 24 hr before recordings, mice were again anesthetized to remove regrowth of the duramater during training. The area around the craniotomy was cleaned with hydrogen peroxide (3%), the silicon elastomer was replaced, and another dose of 10% meloxicam was injected.

### Virus injection & cannula implantation

The surgical procedure was similar to the *head-bar implantation & craniotomy*. The differences in the procedure are outlined below. Bilateral craniotomies were performed at 2.5 mm posterior and 4.3 mm lateral to the bregma. We injected 200 nl AAV5-CAMKII-GtACR2-FRED_kv_2.1, or 200 nl AAV9-CAMKII-dTomato. Both viruses were produced at the ELSC virus-core facility. We injected the virus at a depth of 1.1 mm at 0°. Afterwards, we inserted a 0.2 mm diameter optical fiber with an attached cannula (CFML1202; Thorlabs) at a depth of 0.9 mm at 0°, bilaterally. We chronically fixed the fiber position on the skull using dental cement (Meta bond).

## Auditory stimuli

Sound stimuli were delivered from a calibrated free-field speaker (ES1 SN:4568) using a multifunction processor (ED1; Tucker-Davis Technologies). The speakers were calibrated with a free-field microphone (Type 4939, Bruel and Kjaer). The stimuli were comprised of pure tones, sampled at 500 kHz, between 7.07–14 kHz for behavioral training and 4–40 kHz for passive listening. All stimuli were played at sound pressure levels of 72 dB SPL for behavioral training, and 72-, 62-, 52-, 42-, and 32 dB SPLs for passive listening. All tones were played in randomized order.

## Behavior

### Automated home-cage training (the 'Educage')

For a detailed description of the EduCage behavioral platform, see *Maor et al., 2020*. After RFID implantation, groups of 5 mice were placed in their home cage that was connected to a behavioral chamber called the Educage (*Figure 1a*). Mice were provided with food ad libitum. Water access was restricted to the Educage chamber. Mice could access the Educage freely to retrieve water mixed

with 5% sucrose. The behavioral training was controlled by a custom LabView 2019 (National Instruments) program running on BNC-2110 (National Instruments) and FPGA (MyRIO – National Instruments) to register licks, RFIDs numbers, and deliver stimuli and reinforcements. Every time a mouse entered the water port its RFID was automatically recognized, a trial initiated, and an auditory stimulus played from a speaker vertically positioned above the behavioral chamber. We trained mice gradually for three weeks on a Go/No-Go paradigm of pure tones, spaced around a category boundary of 10 kHz. This paradigm enabled us to efficiently compare between adolescent and adult behavior, since learning was restricted from post-weaning (P20) to adolescence (P37) (correspondingly, adults were trained from P60-P77). Mice were trained in four separate training phases (*Figure 1B*). For the first 24 hr (P20-P21 in the adolescent group of mice and P60-P61 in adult mice) mice were habituated to the behavioral chamber and could approach the port freely to retrieve water after licking the waterspout (0.0015 ml per trial). During the following 48 h (P21-P23 in the adolescent group of mice and P61-P63 in adult mice), mice underwent a tone-association phase. All tones played were 300 ms long (sampled at 500 kHz). Tone association was restricted to the lowest Go tone (7.07 kHz at 100% probability). Mice received a water reward in response to licking during the first 2 seconds after tone offset. This was followed by the first tone-discrimination stage, which contained a No-Go tone (14.14 kHz at 45% probability) at 1 octave away from the Go tone. We analyzed four response outcomes—Hits, Misses, Correct Rejections, False Alarms— in the 2 s response window after the tone-offset. 'Hits' were counted after five successful licks to the Go-tone and followed by a water reward. 'Misses' were counted when licks to the Go-tone trials did not pass the lick threshold. Correspondingly, 'Correct Reject' (CR) trials were all No-Go trials below the lick threshold. CR trials were not reinforced. 'False Alarms' (FA) were counted when 5 or more licks were registered after the No-Go. FA trials were followed by a white-noise punishment (5–20 kHz at 72 dB SPLs, 2 s) and an inter trial delay of 5 s. In addition, mice were exposed to five probe trials (tones whose trial outcome that were neither rewarded nor punished at 8.49 kHz, 9.567 kHz, 10 kHz, 10.44 kHz, and 11.89 kHz; played at 2% per tone). Mice trained on the 1-octave tone discrimination for 1 week (P23-P30 in adolescent mice and P63-P70 in adult mice). Next, we added a second pair of tone - a Go/No-Go tone pair spaced 0.25 octaves apart (9.17 kHz and 10.95 kHz) for another week of training. All tones were played in randomized order, and trial difficulties of the easy and the hard task were intermingled. The experiment was terminated at age P37 for adolescent and P77 for adult mice.

## Head-fixed training

We used a head-fixed paradigm that enabled us to efficiently train adolescent and adult mice on the same task and record from engaged mice after learning. We trained mice on the same Go/No-Go stimulus setup of the Educage described above with few differences which are outlined below.

The head-fixed behavioral training was controlled by a custom MATLAB 2023B (Mathworks) program running on BNC-2110 (National Instruments) to register licks and deliver stimuli and reinforcements. Mice were water-restricted for 24 hr prior to training, and their weight monitored daily. While training on the setup, mice received water supplemented with 5% sucrose, depending on their weight (0.125 ml per g of body weight). Additional water, in the form of Hydrogel (ClearH2O) was given after daily training if mice did not consume the required amount during training, or their body weight dropped below 85% of the pre-restriction weight. Mice were gradually habituated to the setup by head-fixation and received free water after licking, during the first training day (P23 in adolescent mice and P63in adult mice). Licking was measured with an optical lick-meter (Sanworks). During the second training day mice were associated with a tone. All tones of the task were 100 ms long (sampled at 500 kHz). Mice received a water reward in response to licking in the first 2 s after tone offset. Reward reinforcement was delayed to 0.5 s after the tone offset. To break the regularity of the trial sequences, we applied a dynamic inter-trial interval of 6–8 s. New trials were initiated if mice did not lick the spout at least 2 s prior to the upcoming trial. Mice were familiarized with the tone for at least two training days (P24-26 in adolescent mice and P64-66 in adult mice). Then, and when the lick rate was above 80%, mice proceeded to the first tone discrimination stage. We gradually increased the probability of No-Go trials, depending on the lick rate of the mouse. We added the 0.25 octave Go/No-Go tone pair, after a maximal of seven training days on the 1 octave tone pair (P26-33 in adolescent mice and P66-73 in adult mice), or if the behavioral performance exceeded a threshold of d'=1, as calculated using the signal detection metric. Similar to the Educage, all tones were played in randomized order

and trial difficulties of the easy and the hard task were intermingled. Mice were trained until P37 in the adolescent group and P77 in the adult group.

## Optogenetic manipulation

Adult mice (P60) were injected with AAV5-CAMKII-GtACR2-FRED_kv_2.1 (n=3) and implanted with cannulas prior to training as described above. Mice underwent optogenetic stimulation after training on the head-fixed task by attaching a ferrule patch cable (M79L01; Thorlabs) to the implanted cannula using a mating sleeve (ADAF1; Thorlabs). The ferrule cable end (SMA connector) was connected to an LED driver setup (LEDD1B; Thorlabs) that enabled us to send precise outputs at 476 nm (blue light; 5 mW) bilaterally. To test the effect of ACx inhibition on task performance, we timed the LED output to 50 ms before tone onset until 50 ms after tone offset. The LED consisted of pulses given at 10 Hz for 200 ms.

Behavioral sessions of optogenetic manipulation were identical to the Go/No-Go paradigm of previous training sessions. We inhibited ACx in 50% of the trials in randomized order. Optogenetic manipulations were repeated throughout multiple sessions of expert performance. As a control, we repeated the same experiment in mice injected with AAV9-CAMKII-dTomato (n=3). After completion of the experiment, animals were deeply anesthetized with ketamine and medetomidine (0.008 g/kg, and 0.65 mg/kg, respectively) and then perfused with 4% Paraformaldehyde. After the perfusion, the brain was extracted and preserved. We then sectioned the ACx using a cryostat (Leica) into 0.05 mm thick coronal slices. Afterwards, brain slices were washed one time under 1% Phosphate-buffered saline (PBS), and a second time with 1% PBS plus 0.4% Triton. Finally, slides were mounted and stained with DAPI (4%) and imaged using a wide-field fluorescent microscope (Olympus Life Science, Olympus IX83), to reconstruct the probe position offline.

## Extracellular recordings

### Recording setup

Before the onset of the recording session, we removed the silicon elastomer and placed an external reference electrode (Ag/AgCl wire) on the skull of the right hemisphere. All recordings were performed using Neuropixels 1.0 (Npx; IMEC, phase 3 A), together with a base-station connected to a chassis (IMEC; NI PXIe –1071, National Instruments). Probes were mounted to a custom-made steel rod and connected to the ground. Before penetration, probes were covered with a fluorescent dye Dil Invitrogen catalog #V22885- red, or Dio (Invitrogen catalog #V22886 – yellow), to reconstruct penetration sites after the recording.

Then, Npx probes were inserted into the left ACx at 30° to a depth of 3.85 mm (this approach limited our recording to deep layers of the ACx). The skull surface was submerged in saline and the probe was allowed to settle for 10 min. We sampled recordings at 20 kHz, with action potential band filtered to contain 0.3- to 10 kHz frequencies. Action potential band gain was set to 500. Out of the 960 available sites on the 1 cm shank of the Npx probe (*Jun et al., 2017*), we acquired the 384 lowest recording shanks in a staggered configuration. We used common-average referencing to process the voltage traces.

### Recording schedule

We performed recordings of both novice and expert adolescent and adult mice. Novice recordings were performed after tone-association (see *head-fixed training*; adolescent n=3; adult n=3). Expert recordings were performed after learning (adolescent n=5; adult n=6). Both novice and expert recordings include the easy (1 octave) and hard (0.25 octave) tone pair. Both recordings were performed after P37 for adolescent mice and after P77 for adult mice. Tone probabilities and protocol parameters were identical to the *head-fixed training*. Each mouse was recorded multiple times (novice recordings adolescents n=6; novice recording adults n=6; expert recordings in adolescents n=13; expert recordings in adults n=14). In some mice, after the recording in the engaged configuration, we recorded neural activity under passive listening conditions to a pure tone protocol (novice recordings adolescents n=6; novice recording adults n=6; expert recordings in adolescents n=4; expert recordings in adults n=4). The pure tone protocol was comprised of 20 different frequencies, logarithmically spaced between 4 and 40 kHz, presented at five sound pressure levels (see *Auditory stimuli*). Each frequency and attenuation combination were presented 16 times. The tone interval was set to 1 s.

After completion of the recording schedule, animals were deeply anesthetized with ketamine and medetomidine (0.008 g/kg, and 0.65 mg/kg, respectively) and perfused with 4% Paraformaldehyde. After the perfusion, we performed histological analysis and sectioned coronal brain slices as described above (see *Optogenetic Manipulation*) to reconstruct the exact probe position.

### Preprocessing & spike-sorting

All recordings were sorted using Kilosort 2.5/3 open-source software (*Pachitariu et al., 2016*; https://github.com/MouseLand/Kilosort; *Pennington and Pachitariu, 2025*). After automatic sorting, we performed manual curation of the acquired units using 'Phy-2' open-source GUI. (UCL; https://github.com/cortex-lab/phy; *Rossant, 2024*). During manual curation, we distinguished between single units (SUs) and noise (pre-labeled multi-units in Kilosort2.5/3 were automatically labelled as noise). The following parameters were set to determine SUs: physiologically plausible waveform shape (1–2 ms, biphasic/triphasic), high spike-amplitude (>50 μV, SNR >3), physiologically plausible refractory period (ISI >4 ms), sufficient inter-spike-interval (ISI peak 5–10 ms), sufficient firing-rate across the recording (>0.1 spikes/s), and principal component cluster density (high density and low overlap). In addition, SUs were compared to neighboring units, based on waveform, firing rate, drift-pattern, and cross-correlograms to determine merging of two SUs into a single cluster (Pearson Correlation Coefficient >0.8). Units that passed the above-mentioned criteria were considered single units (SUs).

## Data analysis

The analysis was performed with MATLAB R2023b (MathWorks) and Python3.13 (PyCharm 2024.1.3). Violin plots are presented with the median (white dot) together with the interquartile range. Boxplots are presented with the median (horizontal line), mean (gray dot), and quartiles (box). Temporal plots are presented as the mean (line) and the standard error of the mean (patched or error bar).

### Behavioral analysis

We categorized behavioral responses of the task into hit, miss, false alarm, and correct reject responses. To compare behavioral performance, we calculated the d' value based on the signal detection metric (d'=the inverse normal distribution of the z-scored hit rate) – the z-scored (false alarm rate). Z-scored values of 1 were rounded to 0.99, to avoid d' approaching infinity. Unless, defined differently d' was defined for the last 100 trials. Next, we calculated psychometric curves based on the lick rate (above the lick threshold) to learned and probe trials (*Figure 2b, c and e*). Psychometric curves were normalized and fitted to a sigmoidal function, defined as follows:

$$S(t) = a \frac{1}{1 + e^{\frac{-(t-b)}{c}}} \tag{1}$$

Where *a* denotes the lick rate, *b*=the time of the inflection point, and *c*=the steepness of the curve. Based on the normalized psychometric curve, we extracted two inflection points (random lick rate at 0.5 and category boundary at 10 kHz). To compare the heterogeneity of adolescent and adult behavior, we calculated the criterion bias as the inverse normal distribution of the –0.5*(z-scored (hit rate)+z-scored (false alarm rate)) and analyzed the coefficient of variation (CV) of behavior across the learning. We applied similar behavioral measurements to the Educage training and head-fixed training, as well as recordings and optogenetic manipulation during task performance. During expert-performance in the head-fixed task, we analyzed the running d' as the average performance of bins of 25 trials (*Figure 3*). In addition, we also calculated the average lick trace during the reinforcement delay. Here, we also extracted the lick latency per trial per mouse, defined as the lick latency from tone-onset (*Figure 3*).

### Neuronal analysis

Probe trajectories and location of SUs were reconstructed using Allen CCF tools (https://github.com/cortex-lab/allenCCF; *Shamash et al., 2025*; *Shamash et al., 2018*). For further analysis, we only considered SUs that were recorded from infragranular layers 5 and layer 6. SUs from the dorsal-auditory cortex (AUDd), primary auditory cortex (AUDp), ventral auditory cortex (AUDv), and temporal

association cortex (TEa) were pooled together as auditory cortex (ACx), and in some analyses presented separately (*Gilday et al., 2023b*). Neurons were considered excitatory auditory responsive if the average spontaneous firing rate (FR) of the baseline activity preceding the tone (200 ms to 50 ms before stimulus onset) was significantly lower than the average tone-related activity (from stimulus onset until 50 ms after stimulus offset) across all trials (*Feigin et al., 2021*). The difference was tested with a right-sided Wilcoxon signed-rank test (p<0.05). Suppressed auditory responsive units and non-auditory responsive units were excluded from the analysis. Peri-stimulus time histograms (PSTHs) were smoothed with a Gaussian smoothing filter of 5 ms. To test the difference between adolescent and adult auditory firing properties in expert mice, we calculated the difference in average spontaneous FR –200 ms preceding the tone onset until tone-onset, average evoked FR (tone onset until 50 ms after tone offset) and fraction of responsive trials. We also compared the coefficient of variance of the evoked FR (mean FR divided by standard deviation of the FR across trials). Temporal differences in firing properties were compared by calculating the latency to peak (latency of the highest FR), minimal latency (first spike after tone-onset), full-width-half-maximum (FWHM; time from peak FR to baseline FR), and the lifetime sparseness, which was calculated as follows:

$$S = \left(1 - \left(\frac{\left(\frac{\sum r_i}{n}\right)^2}{\frac{\sum r_i^2}{n}}\right)\right) / \left(1 - \frac{1}{n}\right) \quad (2)$$

$r_i$ = corresponds to the FR of each learned frequency and n equals the number of learned tones. Values of S near 0 correspond to a dense FR and values near 1 to a sparse code (*Vinje and Gallant, 2000*).

## Single neuron analysis during task performance

We investigated the FR of excitatory auditory responses per trial from 200 ms preceding the stimulus up to 600 ms after the tone onset. To test how well single neurons discriminate between trial outcomes, as well as task difficulty, we calculated task-related activity per trial using receiver operating characteristics (ROC) and calculated the area under the curve (AUC) across a running window of 25 ms in bins of 50 ms (*Figures 4 and 6*). Stimulus-related activity was defined as the hit compared to the false alarm trials, separately for the 1 octave and 0.25 octave tone pair. Stimulus-related activity of miss compared to correct trials were not analyzed due to the lack of miss trials in expert performance. In addition, we also tested the average (mean of 1 octave and 0.25 octave tone pair) choice discrimination, defined as the difference between false alarm and correct reject trials (*Figure 4*). We sampled 20 trials per trial outcome and repeated the ROC encoding 100 times. Afterwards, we adjusted the average AUC values of all iterations of each neuron to its baseline AUC before the tone onset ((AUC – baseline AUC)+0.5). AUC values close to 0.5 indicate a low neuronal discriminability, and values up to 1 indicate high neuronal discriminability (*Green and Swets, 1966*). To test significant discrimination, we repeated the ROC encoding with shuffled trial identities (20 randomly shuffled trials per trial outcomes across 100 iterations). Significant discriminability of a neuron was defined as the AUC time bins that exceed the mean ± 3 std of the shuffled distribution (*Gilad et al., 2020*). Adolescent and adult neuronal discriminability were compared using the onset-latency of discrimination of a neuron (first timepoint >mean ± 3 std of the shuffled distribution), the maximal AUC of the running window and the duration of discriminability (time duration >mean ± 3 std of the shuffled distribution;). To compare novice and expert performance, we repeated the AUC encoding and compared the general discrimination of 1 octave and 0.25 octave Go and No-Go discrimination of all trial outcomes (*Figure 6*). To evaluate the correlation of behavioral performance and neuronal discriminability, we compared d' per recording to the average of the maximal AUC of all neurons per recording. The AUC and d' values were compared via pairwise Pearson correlation.

## Auditory cortex population analysis during task performance

We analyzed the population activity in the ACx of adolescent and adult mice on a trial-by-trial basis. Population activity was segregated by hit and correct-reject trials for both easy and hard tasks in each age group. We excluded recordings that contained fewer than 20 auditory-responsive neurons, fewer than 20 hit or correct-reject trials in either task. This resulted in a total of 8 adolescent sessions and

10 adult sessions. To decode the stimulus from neural activity, we applied linear-discriminant analysis (LDA) to the first 200 ms following stimulus onset. Decoding accuracy was calculated on held-out trials (10 trials per stimulus; *Figure 5a*). To examine changes in decoding accuracy over time, we used a 50 ms bin width and fitted separate LDA models at each time point, covering a period from –0.5 s to 10 s after stimulus onset (*Figure 5c*). Next, we computed the separation per session, defined as the ratio of the between-class variance to the within-class variance. Given the means $\mu_1, \mu_2$ and variances $\Sigma_1, \Sigma_2$ of the two classes, the separation is given by:

$$S = \frac{\sigma^2_{between}}{\sigma^2_{within}} = \max_w \frac{\left(w\left(\mu_1 - \mu_2\right)\right)^2}{w^T\left(\Sigma_1 + \Sigma_2\right)w}. \tag{3}$$

We used robust linear regression without intercept (Huber loss) to fit the relationship between separations in easy and hard tasks (*Figure 5d*). To quantify when the decoding accuracy is significantly different from the baseline, we used the bin width with a sample frequency of 100 Hz. Baseline decoding accuracy was calculated as the mean and variance of decoding accuracy during a 500 ms window before stimulus onset, up to 50 ms before onset. Decoding latency was defined as the time from stimulus onset to the first time point where decoding accuracy exceeded three standard deviations from the baseline mean (*Figure 5b*). One adolescent session, in which behavioral d' is less than 1, has decoding accuracy that does not exceed three standard deviations from the baseline mean within the first 200 ms. Therefore, this session was excluded from the latency analysis. Finally, we assessed the single-trial variance across both age groups and tasks. Data were standardized per neuron before computing variance per stimulus (*Figure 5e*). To visualize the results, we projected the population representations to the space spanned by the two LDA decoding dimensions for the easy and hard tasks. We averaged the mean and covariance matrix across sessions for each stimulus and age group, representing the distributions with corresponding ellipses (*Figure 5f*).

## Single neuron analysis during passive listening

We extracted the frequency-response areas (FRAs) based on the pure-tone responses (from tone onset up to 50 ms after tone-offset) of auditory responsive neurons. The bandwidth of excited units was computed for significant excitatory responses of adjacent frequencies at 62 dB SPL and subtracted by the number of expected false-positive responses. Afterwards, the sum of significant frequency responses per neuron was multiplied by the octave distance between every frequency to receive an octave-based bandwidth measure. We then calculated the population sparseness as the fraction of significant excited responses in adolescent and adult novice and expert mice. To test the neuronal discriminability to pure tone responses, we calculated the pairwise d' based on the FR per trial in all frequencies at 62 dB SPL (*Feigin et al., 2021*; *Shani-Narkiss et al., 2020*). For two given frequencies p and q, d' values were calculated in the following way:

$$\frac{d\left(\mu_p, \mu_q\right)}{\mu\left(\sqrt{\sum_n^{i=1}\left(\widehat{p}_l - \widehat{q}_l\right)^2}\right)} \tag{4}$$

Where d represents the distance between the mean FR ($\mu$) of frequency p and q, as vectors with n entries representing the mean signal (averaged over individual trials) in n-dimensional space for frequency p/q. The average distance (d) is then divided by the mean of the inner Euclidean distance ($\sqrt{\sum_n^{i=1}(\widehat{p}_l - \widehat{q}_l)}$) between each single trial (*Figure 7h*).

## Statistical analysis

Statistical comparisons were performed in custom-written codes in MATLAB 2023b (MathWorks) and Python 3.12. We assessed if the data was normally distributed using a Kolmogorov-Smirnov test. Normally distributed data was tested using a paired (within group comparison) or independent (between group comparison) t-test and presented with mean ± STE. Non-normally distributed data was tested using a Wilcoxon sign rank (within group comparison), or rank-sum (between group comparison) test. Multiple samples were tested with ANOVA (parametric data) and Kruskal-Wallis tests (non-parametric data). All tests corrected after multiple comparisons (Bonferroni correction for

two-samples and Tukey-Kramer for multiple samples). In addition, we applied linear-mixed-effect models (LME) to account for hierarchical data structures and variability within groups of co-housed mice, as well as repeated measurements of mice and recordings, as random effects. Fixed effects, and pairwise interaction effects were adjusted with post-hoc tests and multiple comparisons. For all statistical tests, significant differences were defined as p-values below 0.05.

## Data availability

The data that support the finding of this study are available through Zenodo (https://zenodo.org/uploads/13933351). Additional data is available from authors upon reasonable request.

## Acknowledgements

We thank the Mizrahi lab for comments on the manuscript and Linda Wilbrecht, and Madeleine Klinger for discussions. We thank Ido Maor, Or Yudco, Omri Gilday, and Meirav Givon for technical assistance. We also thank Ofer Yizhar for sharing reagents and Maya Groysman for preparing the viruses. This work was supported by an NSF-BSF grant to AM and SD (#2021776), stipends from the Minerva Foundation and the Israeli Ministry of Aliya and Integration to BP, and the Gatsby Charitable Foundation. Adi Mizrahi is the Eric Roland Chair in Brain Sciences. Figures 1a and 3a, 4a, 6a, 7a were created with biorender.com.

## Additional information

### Funding

| Funder | Grant reference number | Author |
| --- | --- | --- |
| United States-Israel Binational Science Foundation | 2021776 | Shaul Druckmann<br>Adi Mizrahi |
| National Science Foundation | RO1 DC020874-01 | Shaul Druckmann |

The funders had no role in study design, data collection and interpretation, or the decision to submit the work for publication.

### Author contributions

Benedikt Praegel, Conceptualization, Data curation, Software, Formal analysis, Visualization, Methodology, Writing – original draft; Feng Chen, Formal analysis, Visualization; Adria Dym, Data curation, Methodology; Amichai Lavi-Rudel, Data curation; Shaul Druckmann, Supervision, Funding acquisition, Writing - review and editing; Adi Mizrahi, Conceptualization, Resources, Supervision, Funding acquisition, Methodology, Writing – original draft, Project administration

### Author ORCIDs

Benedikt Praegel http://orcid.org/0009-0006-8253-9625
Feng Chen https://orcid.org/0000-0002-8645-7356
Adi Mizrahi https://orcid.org/0000-0002-1743-6754

### Ethics

All experiments were approved by the Institutional Animal Care and Use Committee (IACUC) at the Hebrew University of Jerusalem, Israel (Permit Number: NS-21-16448-4, NS-21-16694-4, NS-22-16966-4).

Reviewer #1 (Public review): https://doi.org/10.7554/eLife.106387.4.sa1
Reviewer #2 (Public review): https://doi.org/10.7554/eLife.106387.4.sa2
Author response https://doi.org/10.7554/eLife.106387.4.sa3

# Additional files

## Supplementary files

MDAR checklist

Supplementary file 1. Adolescent and adult mice exhibit different lick behavior in the task. Linear mixed-effects models of the fixed effects of lick count (until reward or punishment delay), lick latency, cumulative discriminability (d') (including the interaction effects of lick count and lick latency, lick count and d', lick latency and d', and lick latency, lick count and d') during the minimal number of trials shared between all mice (148 trials; Number of observations = 1098, Fixed effects coefficients = 8, Random effects coefficients = 14, Covariance parameters = 3). Coefficient estimates, STE, T-statistic, degrees of freedom, p-values (adjusted for post-hoc multiple comparisons with Bonferroni method), lower and higher CI are listed in the table. The model includes random effects coefficients per mouse (11 mice in total) and 3 recordings per mouse (see methods, equation 8). Model structure: Lick Count ~ Group * Lick Latency * dprime + (1|Mouse ID) + (1|Recording ID).

Supplementary file 2. Neuronal statistics in expert and novice recordings during task engagement. Acquired single units, acquired tone-excited units (percentage of tone-excited units relative to total units) in the AUDd, AUDp, AUDv, and TEa of adolescent and adult mice in experts (top) and novice (bottom).

Supplementary file 3. Adolescent and adult expert mice have distinct firing properties in different sub-regions. Mean and standard error, mean effect size (robust Cohen's D), lower and upper Confidence Interval (CI) and p-value (Wilcoxon rank-sum test, adjusted for multiple comparisons with Bonferroni method) of the average baseline FR (Hz), evoked FR (Hz), coefficient of variance of FR, latency to peak of maximal FR (ms), full-width-half maximum of peak FR (ms), minimal latency of first spike (ms), fraction of responsive trials, lifetime sparseness of all adolescent and adult neurons from tone-onset to 50 ms after tone offset across all stimuli in AUDd, AUDp, and AUDv, and TEa (significant p-values are highlighted in bold).

Supplementary file 4. Adolescent and adult firing properties of expert mice are distinct between different sub-regions. P-values of Kruskal-Willis Test after Tukey-Kramer correction for multiple comparisons of the average baseline FR (Hz), evoked FR (Hz), coefficient of variance of FR, latency to peak of maximal FR (ms), full-width-half maximum of peak FR (ms), minimal latency of first spike (ms), fraction of responsive trials, lifetime sparseness of all adolescent and adult neurons from tone-onset to 50 ms after tone offset across all stimuli AUDd – AUDp, AUDd – AUDv, AUDd – TEa, AUDp – AUDv, AUDp – Tea, and AUDv –Tea (significant p-values are highlighted in bold).

Supplementary file 5. Novice, adolescent, and adult mice have distinct firing properties in different sub-regions. Mean and standard error (STE), mean effect size (robust Cohen's D), lower and upper Confidence Interval (CI) and p-value (Wilcoxon rank-sum test, adjusted for multiple comparisons with Bonferroni method) of the average baseline FR (Hz), evoked FR (Hz), coefficient of variance of FR, latency to peak of maximal FR (ms), full-width-half maximum of peak FR (ms), minimal latency of first spike (ms), fraction of responsive trials, lifetime sparseness of all adolescent and adult neurons from tone-onset to 50 ms after tone offset across all stimuli in AUDd, AUDp, and AUDv, and TEa (significant p-values are highlighted in bold).

Supplementary file 6. Adolescent and adult firing properties of novice mice are distinct between different sub-regions. P-values of Kruskal-Willis Test after Tukey-Kramer correction for multiple comparisons of the average baseline FR (Hz), evoked FR (Hz), coefficient of variance of FR, latency to peak of maximal FR (ms), full-width-half maximum of peak FR (ms), minimal latency of first spike (ms), fraction of responsive trials, lifetime sparseness of all adolescent and adult neurons from tone-onset to 50 ms after tone offset across all stimuli AUDd – AUDp, AUDd – AUDv, AUDd – TEa, AUDp – AUDv, AUDp – Tea, and AUDv –Tea (significant p-values are highlighted in bold).

Supplementary file 7. Neuronal statistics during passive FRA protocol in expert and novice mice. Acquired single units, acquired tone-modulated units, and percentage of modulated units to all acquired units in the AUDd, AUDp, AUDv, and TEa of adolescent and adult mice during passive-listening recordings.

Supplementary file 8. Adolescent and adult mice have distinct firing properties across different sub-regions of ACx—passive listening. Mean and standard error, mean effect size (robust Cohen's D), lower and upper Confidence Interval (CI) and p-value (Wilcoxon rank-sum test) of the average baseline FR (Hz) (AUDp vs. AUDv: adolescent p = 0.8551; adult p = 0.9711), evoked FR (Hz) (AUDp vs. AUDv: adolescent p = 0.4125; adult p = 0.9954), coefficient of variance of FR (AUDp vs. AUDv: adolescent p = 0.4354; adult p = 0.8800), latency to peak of maximal FR (ms) (AUDp vs. AUDv:

adolescent p = 0.5871; adult p = 0.9985), full-width-half maximum of peak FR (ms) (AUDp vs. AUDv: adolescent p = 0.7223; adult p = 0.4628), minimal latency of first spike (ms) (AUDp vs. AUDv: adolescent p = 0.5936; adult p = 0.5669), fraction of responsive trials (AUDp vs. AUDv: adolescent p = 0.3838; adult p = 0.9924), lifetime sparseness (AUDp vs. AUDv: adolescent p = 0.3792; adult p = 0.9341) of all adolescent and adult neurons from tone-onset to 50 ms after tone offset across all stimuli in AUDp, and AUDv (significant p-values are highlighted in bold).

Supplementary file 9. Overview of datasets analyzed.Number of mice, recordings, and neurons per learning stage (left: expert, right: novice) for every figure.

## Data availability

The data that support the findings of this study are available through Zenodo (https://doi.org/10.5281/zenodo.13933351). The code is available through GitHub (https://github.com/benne1295/Age-and-Learning-Shapes-Sound-Representations-in-Auditory-Cortex-During-Adolescence copy archived at *Praegel, 2025*).

The following dataset was generated:

| Author(s) | Year | Dataset title | Dataset URL | Database and Identifier |
| --- | --- | --- | --- | --- |
| Praegel B | 2025 | Age and Learning Shapes Sound Representations in Auditory Cortex During Adolescence | https://doi.org/10.5281/zenodo.13933351 | Zenodo, 10.5281/zenodo.13933351 |

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
