## [Editor Report · eLife Assessment]

This **important** study suggests that adolescent mice exhibit less accuracy than adult mice in a sound discrimination task when the sound frequencies are very similar. The evidence supporting this observation is **solid** and suggests that it arises from cognitive control differences between adolescent and adult mice. The adolescent period is largely understudied, despite its contribution to shaping the adult brain, which makes this study interesting for a broad range of neuroscientists.

---

## [Referee Report · Reviewer #1 (Public review)]

Summary:

Praegel et al. explore the differences in learning an auditory discrimination task between adolescent and adult mice. Using freely-moving (Educage) and head-fixed paradigms, they compare behavioral performance and neuronal responses over the course of learning. The mice were initially trained for seven days on an easy pure frequency tone Go/No-go task (frequency difference of one octave), followed by seven days of a harder version (frequency difference of 0.25 octave). While adolescents and adults showed similar performance on the easy task, adults performed significantly better on the harder task. Quantifying the lick bias of both groups, the authors then argue that the difference in performance is not due to a difference in perception, but rather to a difference in cognitive control. The authors then used neuropixel recordings across 4 auditory cortical regions to quantify the neuronal activity related to the behavior. At the single cell level, the data shows earlier stimulus-related discrimination for adults compared to adolescents in both the easy and hard tasks. At the neuronal population level, adults displayed a higher decoding accuracy and lower onset latency in the hard task as compared to adolescents. Such differences were not only due to learning, but also to age as concluded from recordings in novice mice. After learning, neuronal tuning properties had changed in adults but not in adolescent. Overall, the differences between adolescent and adult neuronal data correlates with the behavior results in showing that learning a difficult task is more challenging for younger mice.

Strengths:

The behavioral task is well designed, with the comparison of easy and difficult tasks allowing for a refined conclusion regarding learning across age. The experiments with optogenetics and novice mice are completing the research question in a convincing way.

The analysis, including the systematic comparison of task performance across the two age groups, is most interesting and reveals differences in learning (or learning strategies?) that are compelling.

Neuronal recording during both behavioral training and passive sound exposure is particularly powerful, and allows interesting conclusions.

Weaknesses:

The weaknesses listed by this reviewer were addressed by adequate revisions.

---

## [Referee Report · Reviewer #2 (Public review)]

Summary:

The authors aimed to find out how and how well adult and adolescent mice discriminate tones of different frequencies and whether there are differences in processing at the level of the auditory cortex that might explain differences in behavior between the two groups. Adolescent mice were found to be worse at sound frequency discrimination than adult mice. The performance difference between the groups was most pronounced when the sounds are close in frequency and thus difficult to distinguish and could, at least in part, be attributed to the younger mice' inability to withhold licking in no-go trials. By recording the activity of individual neurons in the auditory cortex when mice performed the task or were passively listening as well as in untrained mice the authors identified differences in the way that the adult and adolescent brains encode sounds and the animals' choice that could potentially contribute to the differences in behavior.

Strengths:

The study combines behavioural testing in freely-moving and head-fixed mice, optogenetic manipulation and high density electrophysiological recordings in behaving mice to address important open questions about age differences in sound-guided behavior and sound representation in the auditory cortex.

Weaknesses:

The weaknesses listed by this reviewer were addressed by adequate revisions.

---

## [Author Response]

The following is the authors’ response to the previous reviews.

**Reviewer #1 (Public review):**
Summary:Praegel et al. explore the differences in learning an auditory discrimination task between adolescent and adult mice. Using freely-moving (Educage) and head-fixed paradigms, they compare behavioral performance and neuronal responses over the course of learning. The mice were initially trained for seven days on an easy pure frequency tone Go/No-go task (frequency difference of one octave), followed by seven days of a harder version (frequency difference of 0.25 octave). While adolescents and adults showed similar performance on the easy task, adults performed significantly better on the harder task. Quantifying the lick bias of both groups, the authors then argue that the difference in performance is not due to a difference in perception, but rather to a difference in cognitive control. The authors then used neuropixel recordings across 4 auditory cortical regions to quantify the neuronal activity related to the behavior. At the single cell level, the data shows earlier stimulus-related discrimination for adults compared to adolescents in both the easy and hard tasks. At the neuronal population level, adults displayed a higher decoding accuracy and lower onset latency in the hard task as compared to adolescents. Such differences were not only due to learning, but also to age as concluded from recordings in novice mice. After learning, neuronal tuning properties had changed in adults but not in adolescent. Overall, the differences between adolescent and adult neuronal data correlates with the behavior results in showing that learning a difficult task is more challenging for younger mice.Strengths:The behavioral task is well designed, with the comparison of easy and difficult tasks allowing for a refined conclusion regarding learning across age. The experiments with optogenetics and novice mice are completing the research question in a convincing way.The analysis, including the systematic comparison of task performance across the two age groups, is most interesting, and reveals differences in learning (or learning strategies?) that are compelling.Neuronal recording during both behavioral training and passive sound exposure is particularly powerful, and allows interesting conclusions.Weaknesses:The presentation of the paper must be strengthened. Inconsistencies, missing information or confusing descriptions should be fixed.

We have carefully re-read the manuscript and reviewed it for inconsistencies. We made several corrections in the figures. For example, we removed redundant lines from violin plots and statistics, applied consistent labels, matched y- and x-limits of graphics, and adjusted labels. We also clarified descriptions of some experiment by adding explanations to the text.

The recording electrodes cover regions in the primary and secondary cortices. It is well known that these two regions process sounds quite differently (for example, one has tonotopy, the other not), and separating recordings from both regions is important to conclude anything about sound representations. The authors show that the conclusions are the same across regions for Figure 4, but is it also the case for the subsequent analysis? Comparing to the original manuscript, the authors have now done the analysis for AuDp and AUDv separately, and say that the differences are similar in both regions. The data however shows that this is not the case (Fig S7). And even if it were the case, how would it compatible with the published literature?

To address this and previous concerns about regional differences, the manuscript now includes 4 figures (4-1, 4-3, 6-2, 7-1) and 5 supplemental tables (3,4, 5, 6, 8) that explicitly compare results across brain regions.

Following the reviewer’s request for subsequent analysis, we now added a new supplemental figure (Fig. S6-2) and two new supplementary tables (Tables S5, S6). We show that similar to expert mice (supplementary Table 3, and supplementary Table 4), the firing properties of adolescent and adult novice mice differ across auditory subregions (supplementary Table 5). We also show that the different auditory subregions have different firing properties (supplementary Table 6). With respect to task engagement, we show that (similar to Fig. S4-2) the neuronal discriminability in different auditory subregions is similar in both novice and expert mice (Fig. S6-2).

Following the comment on Fig. S7-1, we made three changes to the revised manuscript. First, we now highlight that the differences firing properties between adolescent and adult neurons in AUDp and AUDv were distinct, but not significantly different within age-group comparisons. Second, we clearly state that the learning related changes in the measured parameters are different between AUDp and AUDv. Note, however, the greater changes in adult neurons after learning remains consistent between AUDp and AUDv. Third, we softened our original claim but still highlighted the stronger learning-induced plasticity in adults.

Regarding the concern that different regions should show different patterns due to their known differences (e.g. tonotopy). Of course we agree that different areas differ functionally (as shown in our own previous work and here as well). However, it is still plausible, and biologically reasonable, that developmental changes may proceed in a similar direction across different areas, even if their baseline coding properties differ.

**Reviewer #2 (Public review):**
Summary:The authors aimed to find out how and how well adult and adolescent mice discriminate tones of different frequencies and whether there are differences in processing at the level of the auditory cortex that might explain differences in behavior between the two groups. Adolescent mice were found to be worse at sound frequency discrimination than adult mice. The performance difference between the groups was most pronounced when the sounds are close in frequency and thus difficult to distinguish and could, at least in part, be attributed to the younger mice' inability to withhold licking in no-go trials. By recording the activity of individual neurons in the auditory cortex when mice performed the task or were passively listening as well as in untrained mice the authors identified differences in the way that the adult and adolescent brains encode sounds and the animals' choice that could potentially contribute to the differences in behavior.Strengths:The study combines behavioural testing in freely-moving and head-fixed mice, optogenetic manipulation and high density electrophysiological recordings in behaving mice to address important open questions about age differences in sound-guided behavior and sound representation in the auditory cortex.Weaknesses:For some of the analyses that the authors conducted it is unclear what the rationale behind them is and, consequently, what conclusion we can draw from them.

We have carefully re-read the manuscript and reviewed it for analyses that lacked a clear rationale or conclusion. To address this, we have made several changes to clarify the reasoning and strengthen the interpretation of the results.

**Reviewer #1 (Recommendations for the authors):**
It would have helped if the authors had highlighted the changes they made to the manuscript compared to the original version - especially since many replies to the reviewers' comments were as vague as "...we fixed some of the wording so it adheres to the data shown", or "we refined our interpretation", without further details.The revised version has improved substantially, and the main claims have been discussed in a more objective way. Important new analyses have been added to allow for a refined interpretation of the results. However, the presentation of the data could still be strengthened significantly (in response to comment A from last review).

We apologize for the lack of detail in some of our previous responses. Our intention was to keep the replies concise, assuming that the side-by-side version with tracked changes would make the edits sufficiently clear. However, we understand the need for greater transparency. Thus, below we provide the following five lists describing the major changes: (1) List of specific reviewer recommendations, (2) list of corrections in figures, (3) list of clarity issues, (4) list of fixed mistakes, (5) list of new figures. We hope this breakdown makes the revisions clearer and more accessible.

List of specific reviewer recommendations:

l.108 mentions a significant change in the vertical line of Fig 1F - Could this significance be indicated and quantified in the figure?

We quantified and indicated the significance of the vertical line in Fig. 1f and Fig. 1i.

Fig.1G - the thick and thin lines should be defined, as well as the grey and white dots (same values for adolescents, not for adults).

(a) We removed the thin inner lines from the violin plot. We define the bar (thick line) of the violin plot in an additional sentence in the methods section under data analysis (LL820-823). (b) We adjusted the marker outlines in the adult data (Fig. 1G).

the figure axis legends should be consistent (trails in Fig D vs # trails in Fig 1F)

We adjusted the axis legend to # trials in Fig. 1D.

l.110: is d' always calculated based on the 100 last trials of a session, or is it just for Figure 1F? -etc...

d’ is always calculated based on the last 100 trials. To clarify this, we added a description in the methods section (L830).

List of corrections in the figures:

(1) We removed the internal lines from violin plots in throughout Fig. 1-7.

(2) We removed the underline of the statistics throughout Fig. 1-7.

(3) We consistently applied ‘adolescent’ and ‘adult’ figure labels and titles with lowercase letters throughout Fig. 1-7.

(4) We applied consistent labelling of ‘time (ms)’ throughout Fig. 1-7.

(5) We matched the size of dashed lines throughout Fig. 1-6.

(6) We adjusted the x-label of Fig. 1d, Fig. S-1-1 a, Fig. 3c, Fig. 3h-i, Fig, 4d to ‘# trials’.

(7) We removed the x-label of ‘Experimental Group’ from Fig. 1 to enhance consistency with other figures.

(8) We removed misaligned dots from the violin plots in Fig. 1g, Fig. 2f, Fig. 3f,g.

(9) We corrected the plot in Fig. S1-1b.

(10) We adjusted the y-limits of Fig. S1-1c to be consistent with Fig. S1-1d,e.

(11) We adjusted the x-labels and y-labels of Fig, 2, Fig. S3-1, Fig, S3-2 and Fig. 3b to ‘freq. (kHz)’.

(12) We added the age of adolescent and adult mice to the schematic timeline in Fig. 2a.

(13) We added a label of the reinforcement delay to the schematic trial structure in Fig. 3b.

(14) We added within-group statistics to Fig. 3e and the figure legend.

(15) We adjusted the x-label of Fig. 3d to ‘# sessions’.

(16) We adjusted the x-label of Fig. 3d and Fig. S3-1b to ‘# licks’.

(17) We changed the y-label in Fig. S3-1a, and Fig. S3-2d, e to ‘lick ratio’ to avoid confusion with the lick rate (Hz) that was calculated in Fig. 4 and Fig. 6.

(18) We replaced the titles ‘CAMKII’ with ‘dTomato’ in Fig. S3-2 to correctly highlight that both the experimental and control injection were CAMKII injections.

(19) We adjusted the x-labels and y-labels of Fig, 2, Fig. S3-1, Fig, S3-2 and Fig. 3b to ‘freq. (kHz)’.

(20) We adjusted the y-label of Fig. S4-1c to ‘# neurons’.

(21) We matched the x-ticks in Fig. 4e,f.

(22) We matched the x-ticks in Fig. 6d-g.

(23) We changed the x-label in Fig. 4g, S4-2 and S6-2 to ‘duration (ms)’ to match the figure label with the manuscript.

(24) We consistently label ‘Hit’, ‘Miss’, ‘FA’ and ‘CR’ with capital letters in Fig. 4d-e.

(25) We replaced the double figure label ‘C.’ in Fig. S4-2 with ‘D.’.

(26) We adjusted the dot-size in Fig. 5 to be equal for all graphs.

(27) We added ticks to the experimental timeline in Fig. 6a.

(28) We corrected the y-label in Fig.7c. Now it correctly reflects 5 attenuations from 72-32 dB SPL.

(29) We matched the y-label of Fig. 7e-h and Fig. S7-1.

List of clarity issues:

(1) We replaced the term ‘lower response bias’ with ‘higher lick bias’ (L24) to accurately describe the more negative (lower) criterion-bias, which highlights a higher tendency to lick.

(2) We replaced the term ‘response bias’ with ‘lick bias’ to consistently describe the calculated criterion-bias (L24, L149, L164, L455, L456, L468).

(3) We clarify that the age-related differences were ‘more pronounced’ instead of simply ‘higher’ to accurately reflect not simply the increase in adolescent lick-bias, but also the decrease in adult lick-bias (L31).

(4) We clarified that adolescent sound representations are not merely ’distinct’, but ‘not fully mature’ in L83.

(5) We clarified in L180 that the impulsive responses we observed in adolescent mice could be related to being ‘less impacted by punishments’.

(6) We clarified the differences in firing properties of auditory sub-regions analyzed in Supplementary Table 3 (L287-295).

(7) We explained and clarified the reference to Fig. 3j (LL252-253).

(8) We added statistics to Fig.S4-2 to support our claim that there are no differences in the onset-latency, duration of discriminability and maximal discriminability between different sub-regions within age-groups (LL 314-315).

(9) We expanded our explanation of the results in Table 3 (LL370-379).

(10) We separated the reference to Fig. 6b and Fig. 6c to clarify their meaning (LL358-361).

(11) We clarified the differences in basic firing properties during the FRA protocol in Fig. 7 (LL409-418).

(12) We expanded our explanation of the differences of the learning related firing properties in AUDp and AUDv of Fig. S7-1 (LL426-433).

(13) We changed the term ‘plasticity profiles’ to ‘learning related plasticity’ to further clarify our limitation that L5/6 and L2/3 may exhibit distinct learning related changes (L496).

(14) We changed the term ‘sluggish’ (L481) to ‘delayed’ to more precisely explain differences between adolescent and adult tuning properties.

(15) We clarified that the running d’ was calculated in bins of 25 trials, instead of ‘the last 25 trials’ (LL845-846).

List of fixed mistakes:

(1) We corrected and matched the age to more accurately reflect the age mice were recorded (P37-42 and P77-82).

(2) We corrected the attenuation range from 72-42 to 72-32 dB SPL to correctly reflect the 5 attenuations used in the protocol.

(3) We corrected the number of channels shown in the voltage trace from 10 to 11 (Fig. S4-1a)

(4) We corrected the number of neurons recorded in novice adolescent mice in the legend of Fig. 6 from 140 to 130 (Fig. 6b).

(5) We removed redundant, or double brackets, commas, dots, and semi-colons in the figure legends.

(6) We corrected the LME statistics Table 2.

List of new figures and tables:

(1) We added a new supplementary figure to accompany Figure 6. Specifically, Fig. S6-2, shows the interaction of the three measured discriminability properties (onset delay, duration of discriminability, and maximal discriminability) in novice compared to expert mice in the easy and hard task (Go compared to No Go). The figure compares the different auditory sub-regions (similar to Fig. S4-2). We show that the discriminability properties within different groups is not significantly different among the four different sub-regions.

(2) Supplementary Table 5: We compared the firing properties in different auditory subregions in novice mice, and found (similar to expert mice) that the firing properties differ between adult and adolescent mice across the four different sub-regions.

(3) Supplementary Table 6: We compared the firing properties between different subregions, separately for adolescent and adult novice mice. Similar to expert mice, we found that different auditory subregions differ in their auditory firing properties.

**Reviewer #2 (Recommendations for the authors):**
The authors largely addressed my suggestions.Comparing hit vs correct rejection trials in the population decoding analysis (L313-314): The authors acknowledge that comparing these two trial types conflates choice and stimulus decoding but I am not convinced that the changes to the manuscript text make this clear enough to the reader.

Thank you for pointing this out. We have made additional revisions to clarify this, and other issues more explicitly, as follows:

(1) We have expanded the explanation of how our population decoding analysis conflates stimulus and choice, and we acknowledge the limitations of this approach in the Abstract (L28), the Results section (L324-326, LL367-370) and the Discussion (LL516-519).

(2) We replaced the analysis of impulsivity on the head-fixed task. Instead of analyzing all it is, we focus only on ITIs following FA trials (Fig. S3-1c,d). This is more consistent with the analysis in the Educage (Fig. S2-1), where we show that adolescents exhibit increased impulsivity after FA trials. We found a similar result for ITIs following FA trials in the head-fixed task.

(3) To provide complementary insight, we now further justify our use of the Fisher separation metric alongside decoding accuracy in Figure 5, with a clearer rationale provided in LL343-345

(4) We also clarified our reasoning for focusing on 62 dB SPL in the FRA-based analysis in LL400-403.